# Massive Sound Embedding Benchmark (MSEB)

**Georg Heigold**
Google Research
Munich, Germany
heigold@google.com

**Ehsan Variani**
Google Research
California, USA
variani@google.com

**Tom Bagby**
Google Research
California, USA
tombagby@google.com

**Cyril Allauzen**
Google Research
New York, USA
allauzen@google.com

**Ji Ma**
Google Research
London, UK
maji@google.com

**Shankar Kumar**
Google Research
New York, USA
shankarkumar@google.com

**Michael Riley**
Google Research
New York, USA
riley@google.com

## Abstract

Audio is a critical component of multimodal perception, and any truly intelligent system must demonstrate a wide range of auditory capabilities. These capabilities include transcription, classification, retrieval, reasoning, segmentation, clustering, reranking, and reconstruction. Fundamentally, each task involves transforming a raw audio signal into a meaningful 'embedding'—be it a single vector, a sequence of continuous or discrete representations, or another structured form—which then serves as the basis for generating the task's final response. To accelerate progress towards robust machine auditory intelligence, we present the Massive Sound Embedding Benchmark (MSEB): an extensible framework designed to evaluate the auditory components of any multimodal system. In its first release, MSEB offers a comprehensive suite of eight core tasks, with more planned for the future, supported by diverse datasets, including the new, large-scale Simple Voice Questions (SVQ) dataset. Our initial experiments establish clear performance headrooms, highlighting the significant opportunity to improve real-world multimodal experiences where audio is a core signal. We encourage the research community to use MSEB to assess their algorithms and contribute to its growth. The library is publicly hosted at https://github.com/google-research/mseb.

## 1 Introduction

Achieving robust machine perception requires more than just seeing; it demands the ability to listen, understand, and act upon the rich acoustic signals in our environment. A comprehensive auditory intelligence must span a diverse array of capabilities, ranging from fundamental perceptual tasks like transcription and segmentation to higher-level cognitive functions such as retrieval, reasoning, and clustering. Traditionally, these tasks have been tackled by disparate research communities, often separated by the domain of sound (e.g., speech vs. non-speech). However, they all share a unified fundamental challenge: transforming high-dimensional raw audio waveforms into meaningful intermediate representations—or 'embeddings'—that can drive downstream reasoning and generation. Recognizing this shared foundation is key to moving beyond narrow, task-specific solutions toward general-purpose multimodal systems.

39th Conference on Neural Information Processing Systems (NeurIPS 2025) Track on Datasets and Benchmarks.

Historically, different applications have evolved specialized forms of embeddings. Technologies involving human speech, like Voice Search [1, 2] or Voice Assistants [3], have focused on creating discrete, textual embeddings through automatic speech recognition (ASR). In contrast, monitoring technologies such as Speaker Verification [4, 5] or general sound classification [6] have concentrated on learning dense, continuous vector representations. While effective for their specific purposes, this fragmentation has made it difficult to assess progress toward a more generalized form of machine auditory intelligence.

Thorough research in this direction requires more than just new models; it demands a standardized way to measure their holistic impact. Specifically, we need to determine: 1) the current state of end-to-end performance at specific compression rates and computational complexities; 2) the maximum performance headroom available; and 3) the existence of general-purpose audio embedding techniques capable of capturing rich acoustic information relevant to multiple tasks. To enable this research, we need a modern benchmark with diverse task coverage, mirroring the role of the massive text embedding benchmark (MTEB) [7] for text and similar benchmarks for images [8, 9]. We introduce the Massive Sound Embedding Benchmark (MSEB) to fill this critical gap, providing a comprehensive forum for the community to compare methodologies and push the boundaries of sound understanding in meaningful, real-world contexts.

Our work makes several key contributions. First, MSEB introduces a suite of eight crucial tasks that cover a wide spectrum of sound-centric technologies, including retrieval, reasoning, and reranking, which have been underrepresented in prior benchmarks. Second, we provide an extensible, open-source library designed to support diverse modeling paradigms relevant to multimodal systems, allowing researchers to seamlessly evaluate their own algorithms. Third, we introduce the Simple Voice Questions (SVQ) dataset, a novel, large-scale resource featuring spoken queries across 26 locales and 17 languages, uniquely designed to support evaluation across multiple tasks from a single source. Finally, our initial experiments establish clear performance headrooms, highlighting the significant opportunity to improve real-world multimodal experiences where audio is a core signal.

MSEB is distinct from prior benchmarks like HARES [10], NOSS [11], and SUPERB [12] in four key ways: 1) it prioritizes tasks with direct real-world technological applicability; 2) it emphasizes measuring embedding efficiency via compression ratio and computational complexity; 3) it includes critical, user-centric tasks such as retrieval and reasoning; and 4) its evaluation methodology does not rely on task-specific fine-tuning, offering a true test of an embedding's generalizable capabilities. We envision MSEB as a dynamic platform and encourage the community to contribute data and tasks to facilitate collaborative advancement in sound technology.

## 2 Massive Sound Embedding Benchmark (MSEB)

The Massive Sound Embedding Benchmark (MSEB) is designed to be the core platform for evaluating the auditory capabilities of any intelligent system, regardless of whether it processes sound as a single modality or as part of a broader multimodal context. MSEB operationalizes these capabilities through defined tasks, carefully selected as verifiable proxies for core challenges in widespread applications. For each task, the model being evaluated must provide a prediction for a given input object—which includes the sound signal and potentially other modalities—drawn from our curated datasets. A fundamental principle of MSEB is verifiability: every task is grounded in established ground truth and a standardized evaluation methodology. To facilitate widespread adoption, the benchmark is supported by a flexible library designed to streamline the evaluation of diverse modeling approaches, from conventional uni-modal models and cascade systems that convert between modalities, to native multimodal models such as Large Language Models (LLMs). While the long-term vision of MSEB encompasses the full spectrum of auditory intelligence, the initial release prioritizes eight tasks with demonstrated real-world utility in rapidly advancing technologies such as voice search, intelligent assistants, and bioacoustic monitoring.

### 2.1 Datasets

A core tenet of MSEB is that robust evaluation requires high-quality, accessible data. We curate our datasets based on five rigorous criteria: (a) public availability with easy-to-use licensing for research; (b) data complexity, ensuring derived tasks are non-trivial and not easily solved by simple baselines; (c) large scale, sufficient to report performance metrics with high statistical confidence;

(d) high-quality labeling, providing verifiable ground truth; and (e) diverse coverage, spanning a reasonable portion of the vast spectrum of sound.

For its initial release, MSEB includes four datasets meeting these criteria, all currently featuring sound and text modalities. We are actively working to expand this coverage to include other modalities, such as images.

**Simple Voice Questions (SVQ)**: We introduce SVQ as a novel, large-scale dataset specifically collected for MSEB. It comprises over 177k short spoken queries across 26 locales and 17 languages, recorded in four distinct environments (clean, background speech, traffic, and media noise). Each query is paired with rich textual metadata derived from the XTREME-UP benchmark [13] (including aligned Wikipedia pages and passages), as well as comprehensive speaker information and fine-grained temporal annotations. Uniquely, SVQ is released as a single, undivided collection. Extensive details on data collection, specific split definitions, and statistical distributions are provided in Appendix A.

**Speech-MASSIVE** [14]: This multilingual Spoken Language Understanding (SLU) dataset extends the text-based MASSIVE corpus [15] with spoken utterances across 12 languages. It features highly granular annotations, covering 18 domains, 60 intents, and 55 slots. We utilize its established test split for standard benchmarking.

**FSD50K** [16]: To cover general sound events, we include FSD50K, containing over 51k clips totaling over 100 hours of audio. It features established development and evaluation splits meticulously designed to prevent data leakage, and uses 200 sound classes drawn directly from the AudioSet ontology [6]. We utilize the standard evaluation split for all MSEB tasks.

**BirdSet** [17]: For bioacoustics, BirdSet provides a massive-scale collection with over 6,800 hours of training data covering nearly 10,000 bird species. It offers seven distinct test sets featuring over 400 hours of soundscape recordings derived from Passive Acoustic Monitoring (PAM) devices. These recordings present realistic challenges like high background noise and overlapping calls. Crucially, they come with strong, temporally precise labels that include rich metadata such as bird type, call type, gender, and geospatial coordinates. MSEB utilizes all seven of these test sets.

## 2.2 Tasks

The initial release of MSEB introduces eight foundational super tasks Figure 1, selected to provide comprehensive coverage of the auditory capabilities expected of a modern intelligent system. To ensure granular and robust assessment, each super task is composed of numerous specific sub-variations, which we refer to simply as tasks. Structurally, every task defines a strict input—comprising the primary audio signal and potentially side information from other modalities—and a required output format. These tasks are selected for their direct alignment with realistic, high-utility applications deployed at scale.

The super tasks are presented below, ordered by a logical progression from user-centric Information Access (Retrieval, Reranking, Reasoning), to fundamental Core Perception (Classification, Transcription, Segmentation), and finally to higher-level Organization  Generation (Clustering, Reconstruction).

**Retrieval** (Finding information): This super task mimics the human process of answering questions by consulting a vast body of knowledge. It serves as a verifiable proxy for Voice Search, where a model is given a spoken query and has access to a "knowledge index"—a bank of information in the form of documents. The task is to find the most relevant information for the query. The target information may be present in the index, or the model may need to correctly determine that no answer is available. We define six sub-tasks based on the granularity of the search and the language match between the query and the index. When searching for a whole document (like a web page) in the same language as the query, the task is DocumentInLang. This is mirrored by PassageInLang (searching for a passage) and SpanInLang (searching for a span of words). Alternatively, a model might hear the question in one language but search across a corpus in another, which defines the DocumentCrossLang, PassageCrossLang, and SpanCrossLang tasks. The SVQ dataset enables evaluation of this super task across all six sub-tasks, 26 language locales, and four different recording environments. It provides a knowledge index (a subset of Wikipedia) and, for each audio query, the corresponding ground-truth pages, passages, and spans.

**Reranking** (Refining information): Intelligent systems, much like humans, often encounter ambiguity where initial perception yields multiple plausible interpretations—for example, when speech is obscured by noise or contains phonetically confused terms. The Reranking super task serves as a proxy for resolving this ambiguity, requiring models to reorder a provided list of candidate hypotheses based on their actual relevance to the original audio signal. This capability is a core component of many practical systems, used to refine candidate documents in search engines or to select the best transcription from an N-best list in automatic speech recognition. In the current MSEB release, we focus on Acoustic Hypothesis Reranking. Leveraging the SVQ dataset, each audio query is paired with a set of text candidates that sound similar to the true utterance. The task is to utilize the audio signal to sort these hypotheses by their true relevance to what was actually spoken.

**Reasoning** (Extracting precise answers): Profound understanding requires more than just locating relevant documents; it demands synthesizing information to derive precise answers. This super task serves as a proxy for next-generation Intelligent Assistants. Assuming relevant knowledge has already been retrieved (e.g., a passage or document), the Reasoning task challenges models to analyze this provided multimodal context in response to a spoken query and pinpoint the exact text span that satisfies the user's information need. Crucially, this requires the robust capability to correctly determine when the provided context contains no answer. In the current MSEB release, leveraging the SVQ dataset, we define two sub-tasks based on language alignment: SpanInLang, where the spoken query and text context share a language, and SpanCrossLang, where they differ.

**Classification** (Identifying sounds): Upon hearing a sound, an intelligent system might need to immediately identify its source, the characteristics of its speaker, or even the intent behind it. This super task evaluates a model's ability to categorize an audio signal into predefined classes based on varied acoustic features. While the potential scope is vast, the initial release of MSEB focuses on four key areas: speaker properties, user intent, environmental sounds, and bioacoustics. For speaker classification (identity, gender, age) and recording environment classification, we utilize the SVQ dataset, which provides rich labels across 26 language locales. Intent classification is evaluated using the Speech-MASSIVE dataset, offering a standard benchmark across 12 languages. For general environmental sound classification, we employ FSD50K with its 200 AudioSet-derived classes. Finally, bioacoustic classification is assessed using the massive-scale BirdSet benchmark.

**Transcription** (Converting sound to text): Upon hearing a sound, a natural response is to transcribe it into a textual representation. While this is most commonly associated with Automatic Speech Recognition (ASR) for spoken language, the concept extends to describing any auditory event in text. This super task evaluates how effectively an intelligent system can translate audio input into a verbatim or descriptive textual output. In the current MSEB release, we focus on ASR, utilizing the SVQ dataset to assess transcription quality across 26 language locales. Furthermore, SVQ's four distinct recording environments allow for a precise comparison of a model's robustness to varied acoustic conditions, such as background noise or media.

**Segmentation** (Locating key information in time): Often, we don't need to hear an entire audio recording; we only need the key information. This super task mimics that need by evaluating a model's ability to identify the most salient moments within an audio signal and pinpoint their precise timestamps. In the current MSEB release, this task is defined using the SVQ dataset across all its 26 language locales. For each audio query, we have identified the top-three most salient terms and their corresponding start and end times. The goal for a model is to correctly predict both these key terms and their exact temporal locations.

**Clustering** (Organizing unknown sound): Beyond just identifying known categories, a truly intelligent system must be able to organize unknown, unstructured sound data into a coherent structure. This super task evaluates a model's ability to discover latent structure by grouping audio segments that share similar properties into homogeneous clusters, without relying on predefined labels. In the initial MSEB release, we define sub-tasks across different domains: speaker, gender, and age clustering using the SVQ dataset; environmental sound clustering using FSD50K; and bioacoustic clustering using BirdSet.

**Reconstruction** (Generating sounds): True mastery of a modality implies the ability not just to understand it, but to generate it. This super task evaluates a system's generative capabilities by requiring it to reconstruct the original audio signal solely from its internal embedding representation. High-fidelity reconstruction serves as a rigorous test that an embedding retains essential acoustic details—beyond just high-level semantics—vital for applications like neural audio compression

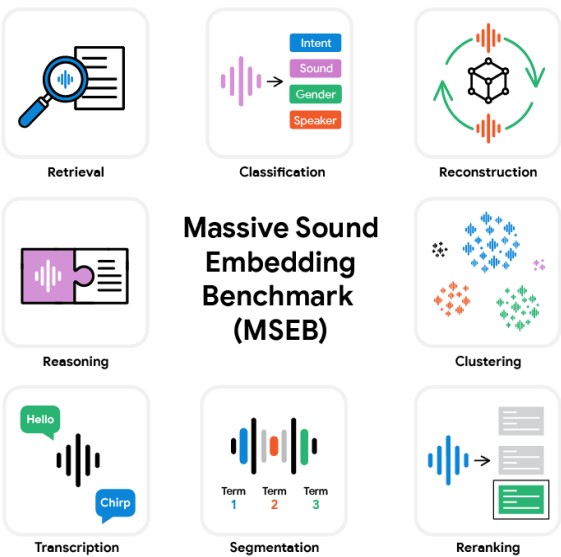

Figure 1: The eight MSEB super tasks, spanning Information Access (Retrieval, Reasoning, Reranking), Core Perception (Transcription, Classification, Segmentation), and Organization Generation (Clustering, Reconstruction).

and speech synthesis. In the initial release, we focus on reconstructing speech in diverse noise environments using the SVQ dataset, alongside general environmental sound reconstruction using FSD50K dataset.

## 2.3 Evaluators

The MSEB evaluator is designed to be strictly model-agnostic. We assume only that a model generates some form of representation for each input example. The evaluator's primary role is to standardise assessment by mapping these representations—optionally, if they are not already in the task's target space—to that space to calculate metrics. While multiple metrics are computed for granular analysis, one primary metric is designated for each task for leaderboard reporting. Beyond task-specific performance, MSEB universally measures efficiency for all entries using two key metrics: Compression Ratio (CR), quantifying the memory footprint of the representation relative to the original audio signal, and Computational Complexity, measured in floating-point operations per second (FLOPS). The following sections detail the specific evaluation protocols for each super task.

**Retrieval**: the evaluator ranks all candidates in the knowledge index based on their proximity to the query in the representation space. The primary metric is Mean Reciprocal Rank (MRR), which rewards models that place the correct target higher in the ranked list. We also report Exact Match (EM), measuring the frequency with which the correct target is the absolute top-ranked candidate.

**Reranking**, the evaluator requires the model to assign a relevance score to each item in a provided list of candidates, which are then sorted by these scores. The primary metric is Mean Average Precision (mAP), assessing the overall quality of the ranked list. We also report Mean Reciprocal Rank (MRR) to measure the rank of the most relevant candidate. Furthermore, to evaluate the utility of the reranking for downstream tasks (e.g., selecting the best ASR transcript), we measure the content quality of the top-ranked candidate using Word Error Rate (WER) and Candidate Error Rate (CER).

**Reasoning**: the model is tasked with extractive question answering, requiring it to identify a precise answer span within a provided context or correctly indicate if no answer exists. The primary performance metric is the gmean-F1 score. This metric measures the overlap between the predicted and ground-truth spans by treating them as bags of tokens, providing a balanced assessment that rewards both exact matches and partial overlaps. Crucially, gmean-F1 is calculated as the geometric mean of the F1 scores for the answerable and unanswerable subsets of the data, rather than the

arithmetic mean (as in vanilla F1). This formulation discourages the trivial strategy of assigning 'No Answer' to all queries while still approximating the vanilla F1 score when performance on both subsets is similar.

**Classification**: the evaluation strategy adapts to the nature of the task's label space. For standard multi-class problems (single label per example), the primary metric is Accuracy. For multi-label problems, where an example may have multiple simultaneous ground truth labels, the primary metric is Mean Average Precision (mAP) (macro-averaged). To ensure a comprehensive assessment, we also report a diverse set of secondary metrics, including top-k accuracy, balanced accuracy, and weighted F1-scores for multi-class tasks, alongside micro/macro F1-scores, Hamming loss, and subset accuracy for multi-label scenarios.

**Transcription**: the evaluator measures the divergence between the model's generated text and the verbatim ground truth. The primary metric is Word Error Rate (WER), a standard measure assessing the proportion of word-level errors (substitutions, deletions, insertions) relative to the reference. To provide finer-grained insights, particularly for languages with complex morphology or to assess spelling accuracy, we also report Character Error Rate (CER).

**Segmentation**: the evaluator assesses the model's ability to identify and localize key events in time. The primary metric is Normalized Discounted Cumulative Gain (NDCG), which evaluates the quality of the predicted sequence by rewarding correct terms found in the correct temporal order. To provide a detailed breakdown of performance, we report three specific accuracy metrics: Content Accuracy (matching the label irrespective of time), Temporal Precision (matching start/end timestamps within a specified tolerance), and the strictest Overall Accuracy (requiring both content and temporal alignment). Additionally, we report Mean Average Precision (mAP) to evaluate detection performance based on confidence scores, and Word Error Rate (WER) to measure overall sequence divergence.

**Clustering**: the evaluator assesses the inherent structure of the embedding space. Given the ground truth number of clusters, we employ a standard MiniBatch K-Means algorithm to group the provided representations. The primary metric is V-measure [18], a score derived from the harmonic mean of homogeneity and completeness, which evaluates how successfully the embeddings separate distinct ground truth categories (e.g., speakers, acoustic environments) into pure and complete clusters.

**Reconstruction**: the evaluator assesses the fidelity of the generated audio by comparing its spectrogram to that of the original reference signal. The primary metric is Fréchet Audio Distance (FAD) [19], which measures the distance between the distributions of the original and reconstructed signal embeddings. To provide a comprehensive view of generation quality, we also report Kernel Audio Distance (KAD) [20], an alternative distribution-based metric using Maximum Mean Discrepancy (MMD), and Embedding MSE, which measures the average per-example mean squared error between the embedding frames.

## 2.4 Library design

MSEB is designed as a flexible library capable of running a diverse set of benchmarks across many types of encoders. The core design principles focus on:

- **Modular Tasks and Encoders**: The library supports modular tasks and encoders that are easy to add and configure.

- **Bulk Encoder Inference**: To accommodate computationally expensive models and large datasets, the library supports bulk inference. Sound datasets often involve significant data sizes and expensive preprocessing (e.g., audio resampling), making efficient bulk processing a critical feature.

- **Diverse Task Types**: MSEB supports a mix of task complexities, ranging from those with expensive setup stages (e.g., retrieval corpus generation) to complex inference steps (e.g., clustering).

The main components and execution flow of the library are illustrated in Figure 2. Task represent instances of a particular benchmark Dataset paired with an Evaluator to score the outputs of an encoder model. The task maps the Dataset (e.g., the Simple Voice Questions dataset) to the multimodal encoder inputs expected by the system (e.g., audio, text, or label pairs).

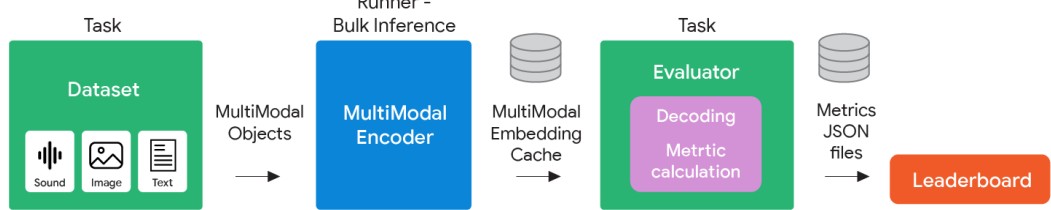

Figure 2: The MSEB library architecture, illustrating the flow from Task Dataset to Leaderboard via bulk inference and evaluation.

For bulk inference, the task explicitly exposes the set of inputs that need to be encoded. A `Runner` interface exists to execute the encoder over this set in various ways, such as through a beam pipeline. The output of the runner is a key-value store (the Embedding Cache) mapping unique identifiers used by the task to the encoder outputs. This design allows separating bulk encoding steps into multiple stages with different resource requirements.

Encoder models implement a `MultiModalEncoder` interface, which supports an extensible set of input and output types with runtime checks. While currently oriented around Sound and Text modalities, it is designed to be extensible to others, such as images, in future iterations. The definition of an encoder output encompasses many styles of "embedding": fixed-size vectors, sequences of embeddings, or discrete tokens.

Encoders can also be combined; we currently support three main types:

1. **End-to-End Encoder**: Takes multimodal input and directly produces an embedding.
2. **Cascade**: A sequence of encoders chained together (e.g., ASR → text embedding).
3. **Collections**: Multi-tower encoders or combinations of encoders for different modalities.

The Evaluator handles the correct usage of these diverse encoder outputs to compute metrics. For instance, in classification, an end-to-end encoder might output a class label text token directly, while a fixed-size vector encoder might need to be paired with label vectors and evaluated using cosine distance.

The core function of the library is to run a benchmark instance of a (`task, encoder`) pair. Registries are provided for listing and selecting these instances. Each benchmark run generates a JSON file containing complete metadata and metrics. Leaderboard submissions are facilitated by submitting pull requests to add these output JSON metric files, which are then collated into the final display.

## 3  Experiments

We utilized the MSEB framework to establish initial standard baselines and quantify the performance "headroom" across all eight super tasks. These widespread experiments provide a unified view of the current state of machine auditory capabilities, highlighting both existing strengths and significant opportunities for improvement.

Figure 1 presents a comprehensive overview of these results. In this figure, each bar represents the primary metric for a specific task, identified in brackets next to the task name. The bars range from 0 to the metric's maximum value. For bounded metrics (e.g., Accuracy, F1), the bar ends at 1.0. For unbounded metrics (e.g., WER, FAD), a dashed line at the end indicates their open-ended nature, with each internal bin representing one-tenth of the displayed range (the scale for these is noted next to the bar). Furthermore, each bar visualizes performance variability: the labels indicate the specific sound types (e.g., language locales or domains) that achieved the minimum and maximum scores, while the gradient color between them represents the distribution of all other sound types relevant to that task.

To quantify this performance gap, we adopted a unified comparison where applicable. Green bars show performance when sound was used as the primary input to the task, representing typical current systems. Blue bars show performance when the corresponding ground-truth text transcript was

used instead, representing an "oracle" scenario with perfect perception. The gap between these bars quantifies the immediate headroom available for improving auditory intelligence, particularly for tasks that mainly rely on the semantic content of the sound rather than its other acoustic characteristics.

Two main observations can be drawn from this overview: a) significant opportunity for improvement exists across all tasks to reach the maximum possible score, and b) current sound-based approaches significantly lag behind those using the text modality, suggesting that more research is needed to design better sound representations for robust auditory understanding.

Next, we provide further details on the specific baselines evaluated for each super task. For comprehensive comparisons across all sub-tasks, including secondary metrics and the latest submissions, we refer readers to the live MSEB leaderboard and the detailed appendices.

**Retrieval**: For this super task, we adopted a cascade encoding approach, which remains the state-of-the-art for current industrial voice search systems [21]. Our standard baseline (Sound Input) utilized Whisper Large v3 [22] for automatic speech recognition (ASR), followed by a text embedding model to encode the resulting transcription. We evaluated two distinct text encoders: GeminiEmbedding [23], chosen for its state-of-the-art performance on the MTEB leaderboard at the time of our experiments, and Gecko [24], selected to provide a comparative baseline using a smaller, less powerful text model.

Evaluation was conducted across all SVQ locales. For each example, the spoken query was transcribed and then embedded. Simultaneously, the document index was explicitly encoded using the same text embedding model. Retrieval was performed via dot product between the query embedding and document embeddings, with quality measured by Mean Reciprocal Rank (MRR).

To establish the performance headroom (Text Input in Figure 1), we conducted an "oracle" experiment where ground-truth human transcriptions were fed directly to the text encoders, bypassing the ASR step.

Our primary observations are threefold. First, as shown in Figure 1, sound-based models consistently lag behind their text-only counterparts across all languages, establishing a clear gap for future end-to-end audio models to close. Second, ASR quality (WER) does not perfectly correlate with retrieval performance (MRR) across different languages (detailed in Appendix B). This likely arises because standard ASR objectives treat all words equally, whereas effective retrieval depends disproportionately on semantically salient terms. Third, comparing in-language and cross-language tasks reveals a clear performance gap for both sound and text modalities. This suggests that current embedding vectors have not yet fully achieved universal semantic representation and remain bound by language. Interestingly, this gap is relatively smaller for passage retrieval compared to full-page retrieval, implying that shorter document contexts may facilitate better cross-lingual alignment.

**Reranking**: For this task, we focused on Acoustic Hypothesis Reranking using the SVQ dataset. Our baseline (Sound Input) employed the same cascade approach as used in Retrieval: audio queries were transcribed by Whisper Large v3, and both these transcripts and the provided candidate hypotheses were embedded using GeminiEmbedding. Candidates were then re-ordered based on their cosine similarity to the query's embedding. The Text Input headroom was established by using the ground-truth transcript as the query for reranking.

Results show extreme variability across locales, heavily correlated with the underlying ASR quality. While English locales achieve high performance (e.g., en-AU at 0.86 mAP), challenging languages like Malayalam (ml-IN) see severe degradation (down to 0.11 mAP). This confirms that current text-based reranking is heavily bottlenecked by the initial 1-best ASR output. A direct sound-based reranker that leverages acoustic nuances lost in transcription could potentially bypass this limitation.

**Reasoning**: We employed the Gemma 3 foundational model [25] for this task, prompting it to extract precise answer spans from the provided text context. We compared a Sound Input baseline (feeding Whisper ASR transcripts to Gemma) against a Text Input oracle (feeding ground-truth transcripts).

Results indicate significant headroom, with the performance gap heavily dependent on the language and task complexity. While some languages like Arabic show resilience, others suffer dramatic performance losses due to ASR errors. For instance, in Cross-Language reasoning for Malayalam (ml-IN), F1 scores plummet from 0.54 (Text) to just 0.12 (Sound). Even in simpler In-Language scenarios, clear gaps persist (e.g., Bengali drops from 0.65 to 0.54 F1), demonstrating that current ASR systems often fail to preserve the precise semantic details necessary for robust downstream reasoning.

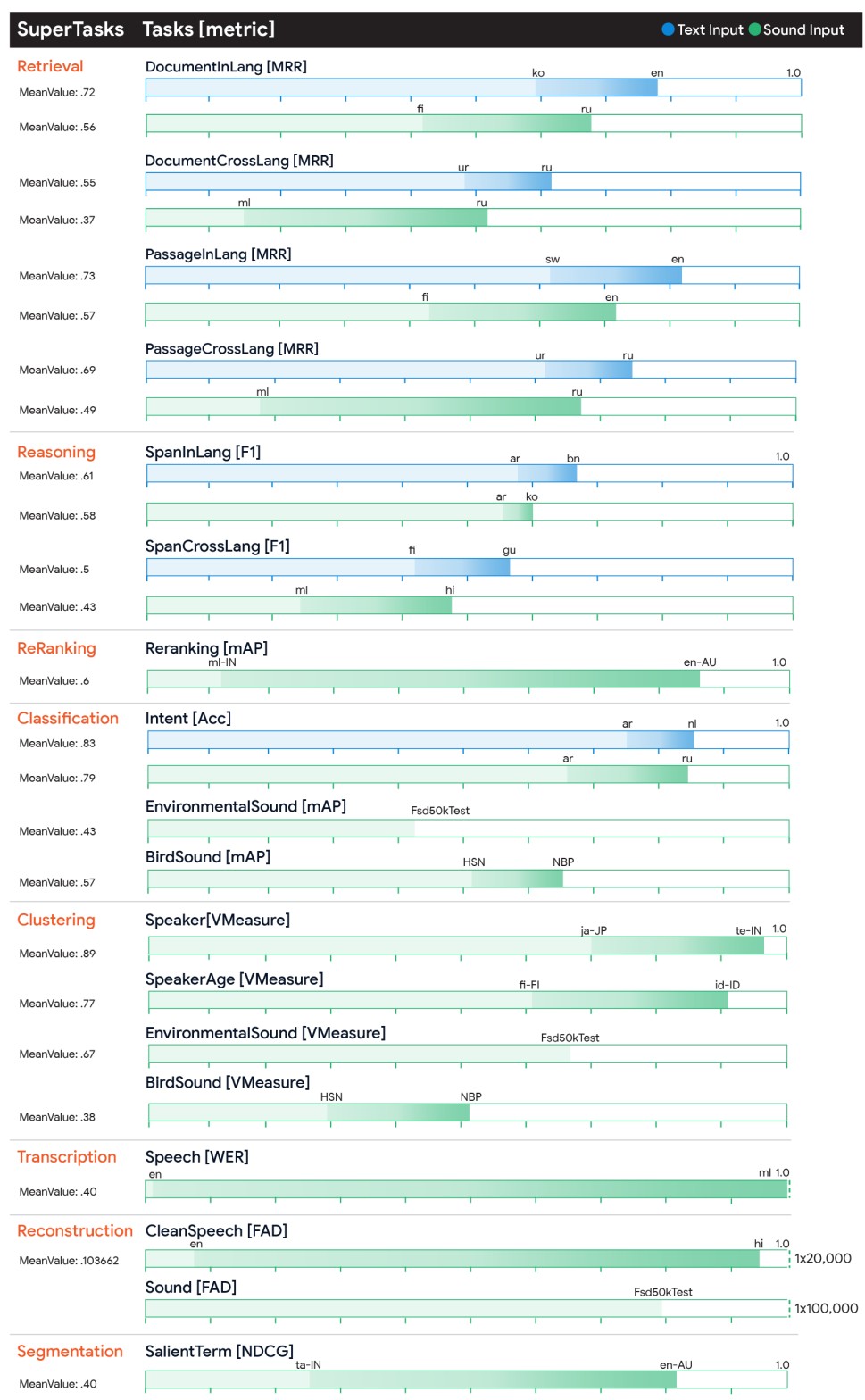

Figure 3: Performance overview of the eight MSEB super tasks. Green bars show performance when sound was used as input to the task, while blue bars show performance using the corresponding ground-truth text transcript. Range markers indicate variability across locales or domains.

**Classification**: We evaluated diverse modeling strategies across different domains for this task. For Intent Classification on Speech-MASSIVE, we employed the same cascade baseline as in Retrieval (Whisper ASR → GeminiEmbedding), where predictions were made by finding the class label embedding with the smallest cosine distance to the input embedding. Comparing Sound Input (cascade) against Text Input (oracle transcriptions) reveals varying degrees of headroom; while some languages like German show minimal loss (0.84 to 0.83 accuracy), others like Arabic suffer more significant degradation (0.77 to 0.67), highlighting ASR's varying impact on downstream understanding.Conversely, for non-speech tasks, we utilized direct audio encoders to demonstrate MSEB's breadth. For Environmental Sound Classification on FSD50K, we evaluated the CLAP encoder [26], leveraging its dual-encoder architecture to perform zero-shot classification by matching audio embeddings directly to class label text embeddings (achieving 0.43 mAP). For Bioacoustics on BirdSet, we evaluated Perch [27], a specialized wildlife audio model, using its direct logit outputs (achieving, for example, 0.66 mAP on the NBP test set).

**Transcription**: We evaluated the robustness of state-of-the-art ASR by transcribing all 26 SVQ locales using Whisper Large v3. Results show significant variability across languages, with Word Error Rates (WER) ranging from as low as 7.7% for Arabic in clean conditions to over 100% for Malayalam, indicating severe challenges for some long-tail languages even with powerful models. Furthermore, environmental noise has a measurable but varied impact; for example, English WER degrades from 1.6% in clean conditions to 3.8% in background speech, while other languages show different sensitivities to specific noise types.

**Segmentation**: We established a baseline for this task using a cascade approach on the SVQ dataset. Audio was transcribed by Whisper Large v3, and the top-three salient terms were identified using a predefined IDF table. We then utilized Whisper's inherent word-level timestamps to localize these terms. Performance, measured by NDCG, is highly variable and inextricably linked to ASR quality. While English locales achieve high precision (e.g., en-AU at 0.83), performance collapses for challenging languages like Bengali (0.05) and Malayalam (0.002), illustrating the brittleness of relying entirely on explicit text transcription for localizing semantic content.

**Clustering**: We evaluated discovering latent structure across diverse domains using domain-appropriate encoders. For Bioacoustics (BirdSet), Perch embeddings [27] proved highly effective, achieving V-measures up to 0.53 on complex soundscapes (NBP), significantly outperforming general audio baselines. For Environmental Sounds (FSD50K), the CLAP encoder [26] demonstrated strong performance (V-measure 0.68), successfully organizing diverse audio events without explicit labels. In contrast, for Speaker Clustering on SVQ, simple raw spectrogram features surprisingly matched or even outperformed sophisticated pre-trained models like HuBERT [28] and Wav2Vec2 [29] in many locales (e.g., achieving 0.97 V-measure for speaker ID in te-IN), suggesting that fundamental acoustic properties are often sufficient for basic speaker discrimination in clean conditions.

**Reconstruction**: To establish a baseline for this task, we evaluated EnCodec [30], a widely adopted neural audio codec. We measured the fidelity of the reconstructed audio against the original signal utilizing Fréchet Audio Distance (FAD) on both SVQ and FSD50K.Results indicate that current models are highly optimized for specific languages and conditions but lack universal robustness. On SVQ, even for clean speech, performance varies drastically by locale, ranging from ∼15k FAD for English to over 190k for Hindi. Quality degrades further in noisy environments (e.g., English in traffic noise jumps to ∼490k FAD). General environmental sound reconstruction on FSD50K proves even more challenging, yielding the highest FAD score (∼778k) and highlighting the substantial need for improved universal audio generative models.

## 4    Conclusion

he Massive Sound Embedding Benchmark (MSEB) aims to be the definitive platform for evaluating the sound capabilities of intelligent systems. Its initial release, featuring eight diverse super tasks and four large-scale datasets, covers a significant portion of the sound spectrum and essential real-world functionalities. MSEB's flexible, model-agnostic library enables seamless evaluation of any architecture, from conventional models to LLMs. Our initial experiments reveal substantial performance headrooms between current sound-based methods and text-based oracles, alongside significant variability across different sound types, languages, and noise conditions. This highlights the urgent need for robust, universal sound representations. We invite the research community to contribute to MSEB, fostering collaborative progress toward truly general machine sound intelligence.

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

## A SVQ Dataset

The SVQ dataset is an open-source collection, accessible at `https://huggingface.co/datasets/google/svq`, which was specifically curated to support a variety of tasks within the MSEB benchmark. This section provides a high-level statistical overview and descriptive details of this dataset.

**Splits**: The audio data in this release is presented as a single, comprehensive collection, rather than being pre-divided into training, validation, or testing subsets. This decision stems directly from the design of the data acquisition process. Specifically, text prompts and recording environments were randomly allocated across the speaker cohort. While this approach promotes a rich variety of conditions, it introduces a complexity for traditional data splitting: creating partitions that ensure no overlap of speakers and no overlap of text material between splits (a common best practice) would lead to a substantial data reduction, estimated at around 40% of the total recordings.

The primary goal guiding this release strategy is to maximize the utility and volume of the data available to users. Therefore, to avoid this significant data loss and provide the fullest possible dataset, the data is released in its entirety as an undivided evaluation set. Users intending to train models with this data will need to devise and implement their own splitting strategies, keeping in mind the inherent trade-offs between data volume and strict speaker/text disjointness if they attempt to replicate such conditions.

**Linguistic Composition**: The dataset includes 26 language locales, representing 17 different languages. Languages, sourced in correspondence with the XTREME-UP benchmark [13], were recorded across 1 to 5 locales each to ensure representation of various regional dialects and accents. Recordings from all identified locales are incorporated into the final dataset. Languages with multiple locales include:

- English (en): en-AU, en-GB, en-IN, en-PH, en-US

- Arabic (ar): ar-EG, ar-x-gulf, ar-x-levant, ar-x-maghrebi

- Bengali (bn): bn-BD, bn-IN

- Urdu (ur): ur-IN, ur-PK

Note: uk-PK in the original text is presumed to be ur-PK for consistency with the language code.

**Dataset Scale and Audio Characteristics**: Mean recording duration: 5.1 seconds
Minimum recording duration: 1 second
Maximum recording duration: 62.4 seconds
Total individual recordings: 171,434
Distinct text transcripts (prompts): 25,549
Total unique speakers: 700

An overview of the SVQ dataset's high-level statistics is presented in Table 1.

**Speaker information**: The SVQ dataset offers valuable speaker-specific metadata, including anonymous identifiers, self-reported gender, and age, for its 700 unique participants. An examination of the gender distribution reveals a diverse composition: the predominant group consists of 376 speakers identifying as female (approximately 53.7%), followed by 308 speakers identifying as male (approximately 44.0%). Additionally, the dataset includes a smaller segment of 2 individuals identifying as non-binary (around 0.3%) and 14 speakers (2.0%) who opted not to provide an answer regarding their gender. Collectively, these figures depict a generally balanced representation between the two largest gender categories, albeit with a slightly greater prevalence of female participants. For a more detailed, locale-specific breakdown of this gender distribution, refer to Table 2.

The overall age profile of the SVQ dataset, as detailed in Table 3, indicates a participant pool primarily composed of adults. Across all locales, the median age of speakers is 29 years. The full age spectrum represented in the dataset is quite broad, with the youngest participant being 18 years old and the oldest being 71 years old. This range suggests that while the central tendency leans towards young to

Table 1: SVQ dataset: general statistics.

| Language | Recordings | Speakers | Locale | Recordings | Speakers |
|---|---|---|---|---|---|
| ar | 52453 | 229 | ar-EG | 13811 | 59 |
| | | | ar-x-gulf | 13452 | 58 |
| | | | ar-x-levant | 12600 | 55 |
| | | | ar-x-maghrebi | 12590 | 57 |
| bn | 8845 | 42 | bn-BD | 4119 | 22 |
| | | | bn-IN | 4726 | 20 |
| en | 28004 | 121 | en-AU | 5506 | 25 |
| | | | en-GB | 5586 | 25 |
| | | | en-IN | 5632 | 23 |
| | | | en-PH | 5647 | 24 |
| | | | en-US | 5633 | 24 |
| fi | 10365 | 51 | fi-FI | 10365 | 51 |
| gu | 3689 | 16 | gu-IN | 3689 | 16 |
| hi | 3805 | 18 | hi-IN | 3805 | 18 |
| id | 5726 | 28 | id-ID | 5726 | 28 |
| ja | 2833 | 13 | ja-JP | 2833 | 13 |
| kn | 3453 | 15 | kn-IN | 3453 | 15 |
| ko | 6520 | 28 | ko-KR | 6520 | 28 |
| ml | 3353 | 16 | ml-IN | 3353 | 16 |
| mr | 3699 | 16 | mr-IN | 3699 | 16 |
| ru | 11912 | 48 | ru-RU | 11912 | 48 |
| sw | 5878 | 25 | sw | 5878 | 25 |
| ta | 3308 | 16 | ta-IN | 3308 | 16 |
| te | 10410 | 45 | te-IN | 10410 | 45 |
| ur | 7181 | 32 | ur-IN | 3688 | 16 |
| | | | ur-PK | 3493 | 16 |

middle-aged adults, there is also inclusion of both younger adults at the cusp of adulthood and more senior individuals.

Examining the age distributions within specific language locales reveals considerable variability, highlighting the diverse demographic makeup of the dataset. Median ages at the locale level fluctuate, ranging from a younger median of 24 years for speakers in the fi-FI (Finnish in Finland) locale, to notably older medians such as 39 years for en-AU (English in Australia), and 37 years for both ru-RU (Russian in Russia) and ja-JP (Japanese in Japan). This indicates that the typical age of participants can differ substantially from one region or language group to another.

Furthermore, the age spans (from minimum to maximum age) within locales also show diversity. For instance, the ru-RU locale includes speakers up to 71 years old, mirroring the overall maximum age in the dataset, and fi-FI includes speakers up to 67 years. In contrast, some locales exhibit a more constrained age range, such as gu-IN (Gujarati in India), where the maximum participant age is 34, and bn-BD (Bengali in Bangladesh) with a maximum age of 36. This variation underscores that while some locales capture a very wide age demographic, others focus on a more specific, often younger, adult age group.

Table 2: SVQ dataset: speaker gender statistics.

| Language | Locale | Female | Male | No Answer | Non-Binary |
|----------|--------|--------|------|-----------|------------|
| ar | ar-EG | 28 | 31 | | |
| | ar-x-gulf | 38 | 16 | 4 | |
| | ar-x-levant | 36 | 16 | 3 | |
| | ar-x-maghrebi | 28 | 29 | | |
| en | en-AU | 16 | 9 | | |
| | en-GB | 15 | 9 | | 1 |
| | en-IN | 12 | 8 | 3 | |
| | en-PH | 20 | 4 | | |
| | en-US | 10 | 12 | 1 | 1 |
| bn | bn-BD | 4 | 18 | | |
| | bn-IN | 4 | 16 | | |
| fi | fi-FI | 12 | 38 | 1 | |
| gu | gu-IN | 8 | 8 | | |
| hi | hi-IN | 9 | 8 | 1 | |
| id | id-ID | 23 | 5 | | |
| ja | ja-JP | 8 | 4 | 1 | |
| kn | kn-IN | 11 | 3 | 1 | |
| ko | ko-KR | 19 | 9 | | |
| ml | ml-IN | 9 | 7 | | |
| mr | mr-IN | 8 | 7 | 1 | |
| ru | ru-RU | 31 | 16 | | 1 |
| sw | sw | 12 | 13 | | |
| ta | ta-IN | 10 | 6 | | |
| te | te-IN | 18 | 27 | | |
| ur | ur-IN | 6 | 10 | | |
| | ur-PK | 9 | 7 | | |
| total | | 376 | 308 | 14 | 2 |

Table 3: SVQ dataset: speaker age statistics.

| Language | Locale | Median Age | Min Age | Max Age |
| --- | --- | --- | --- | --- |
| ar | ar-EG | 32 | 20 | 56 |
| | ar-x-gulf | 30 | 19 | 48 |
| | ar-x-levant | 30 | 20 | 52 |
| | ar-x-maghrebi | 27 | 18 | 46 |
| en | en-AU | 39 | 22 | 65 |
| | en-GB | 27 | 20 | 49 |
| | en-IN | 27 | 21 | 46 |
| | en-PH | 31 | 20 | 39 |
| | en-US | 28 | 20 | 62 |
| bn | bn-BD | 28 | 21 | 36 |
| | bn-IN | 30 | 24 | 51 |
| fi | fi-FI | 24 | 20 | 67 |
| gu | gu-IN | 28 | 23 | 34 |
| hi | hi-IN | 31 | 19 | 47 |
| id | id-ID | 28 | 22 | 46 |
| ja | ja-JP | 37 | 28 | 52 |
| kn | kn-IN | 28 | 22 | 49 |
| ko | ko-KR | 35 | 20 | 55 |
| ml | ml-IN | 28 | 22 | 41 |
| mr | mr-IN | 28 | 22 | 45 |
| ru | ru-RU | 37 | 21 | 71 |
| sw | sw | 27 | 22 | 40 |
| ta | ta-IN | 32 | 22 | 48 |
| te | te-IN | 26 | 19 | 46 |
| | ur-IN | 31 | 21 | 4 |
| ur | ur-PK | 32 | 22 | 43 |
| total | | 29 | 18 | 71 |

# B Retrieval

We define retrieval tasks at three distinct levels of granularity: page retrieval, passage retrieval, and span retrieval. Each of these tasks is addressed in two variants: in-language and cross-language retrieval.

For passage retrieval, we utilize the indexes provided by XTREME-UP. A single index containing 271,711 passages, including those from 9 languages and additional hard negatives, supports all in-language retrieval tasks. For cross-language retrieval, we use a separate index of 112,426 passages from the English Wikipedia.

For page retrieval, we constructed indexes in two sizes: 'small' and 'full'. The 'small' in-language and cross-language page indexes are created by mapping passages from the previously described XTREME-UP passage indexes to their corresponding page titles, followed by deduplication. The pages referenced by these titles are drawn from the wikipedia/20190301.[ar|bn|en|fi|id|ko|ru|sw|te] TensorFlow Datasets (TFDS) snapshot[1]. This process yields 8,568 distinct pages for the 'small' in-language index (aggregating these nine languages) and 4,559 for the 'small' cross-language index (derived from English page titles).

In contrast, the 'full' indexes utilize all pages from each available language edition within the wikipedia/20190301.<lang> TFDS snapshot. Unlike the multilingual 'small' in-language page index, the 'full' in-language page indexes are language-specific. Furthermore, the 'full' English (en) index serves as the 'full' cross-language retrieval index. The sizes of these 'full' indexes vary significantly, ranging from several thousand to several million pages per language, thereby spanning three orders of magnitude, see Table 4.

Table 4: MSEB page index sizes (full).

| ar | bn | en | fi | id | ko | ru | sw | te |
|---|---|---|---|---|---|---|---|---|
| 5,824,596 | 1,272,226 | 87,566 | 619,207 | 947,627 | 980,493 | 2,449,364 | 48,434 | 91,857 |

The textual content of pages is substantially greater than that of passages and can thus exceed the maximum context length of the employed embedding models. For instance, the Gecko and Gemini models impose an 8,000-token limit. Fig. 4 illustrates the page text length distribution for a selection of languages. As depicted, while most pages (say, 99%) are shorter than this limit, a small portion is considerably longer. Where page length surpasses this capacity, we segment the text into chunks and utilize the mean of their respective embeddings as the page's final representation.

For span retrieval, indexes are constructed from the corresponding passage-level indexes, using a methodology similar to that for the 'small' page retrieval indexes detailed previously. We recommend limiting spans to a maximum of ten tokens.

Comprehensive per-language results for cascade models, including headroom analysis using ground truth transcripts, are presented in Fig. 6 (Gecko) and Fig. 7 (Gemini Embedding). We observe that the headroom varies across retrieval tasks based on factors such as language, in- vs. cross-lingual conditions, and index size.

We highlight the following key observations:

- Whisper demonstrates significant performance differences across languages (see Fig. 5), with word error rates (WERs) ranging from $10\%$ (en) to $100\%$ (ml).

- Both Gecko and Gemini Embedding exhibit consistent performance across (most) languages.

- Gemini Embedding consistently outperforms Gecko.

- Notably, headroom persists even with nearly 'perfect' text embedders because answers cannot always be reliably inferred from noisy text, as exemplified by page retrieval tasks using the 'small' index.

- Conversely, headroom tends to be limited when text embedder performance is low, which is evident in cross-language page retrieval tasks with the 'full' index.

---

[1] https://www.tensorflow.org/datasets/catalog/wikipedia

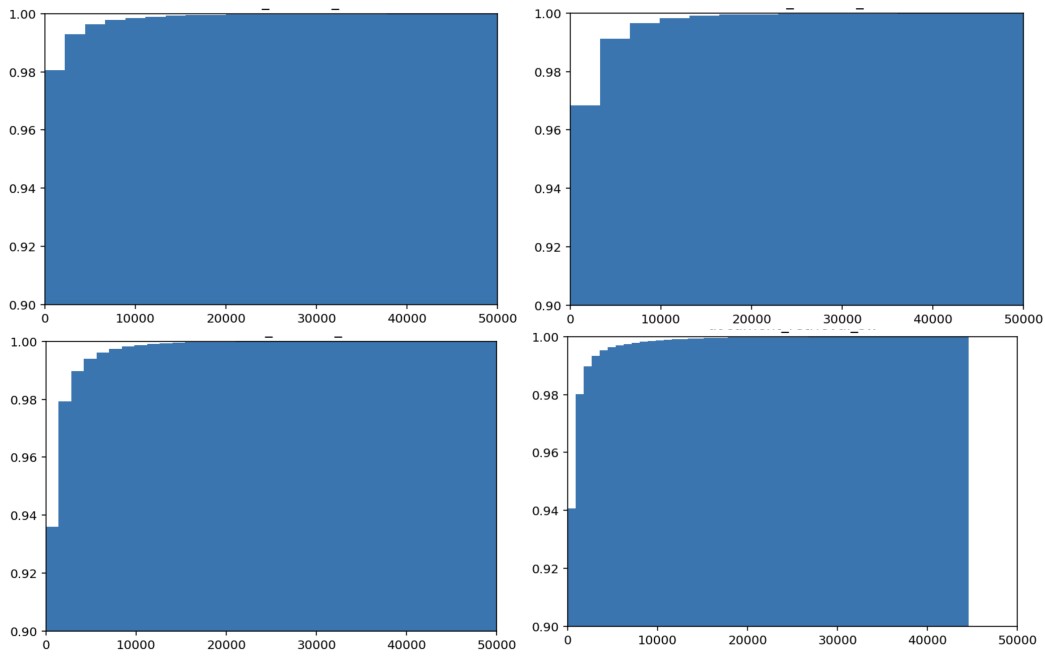

Figure 4: Typical page length distributions (in Gemini Embedding tokens).

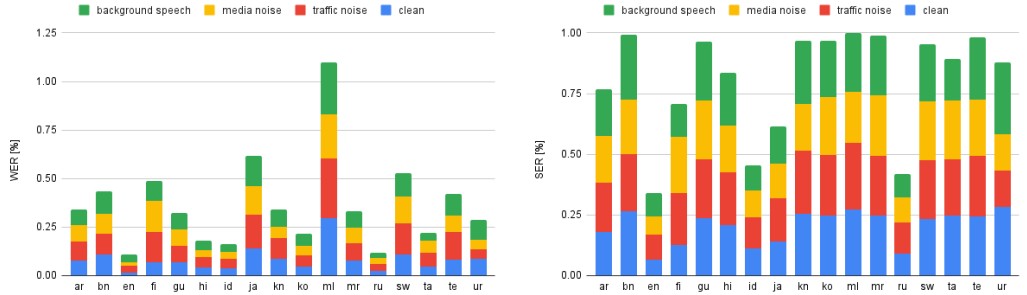

Figure 5: Whisper word error rates (left) and sentence/query error rates (right) across different languages and environments.

Overall, a substantial headroom exists between ASR-based models and ground truth, indicating considerable potential for improved modeling.

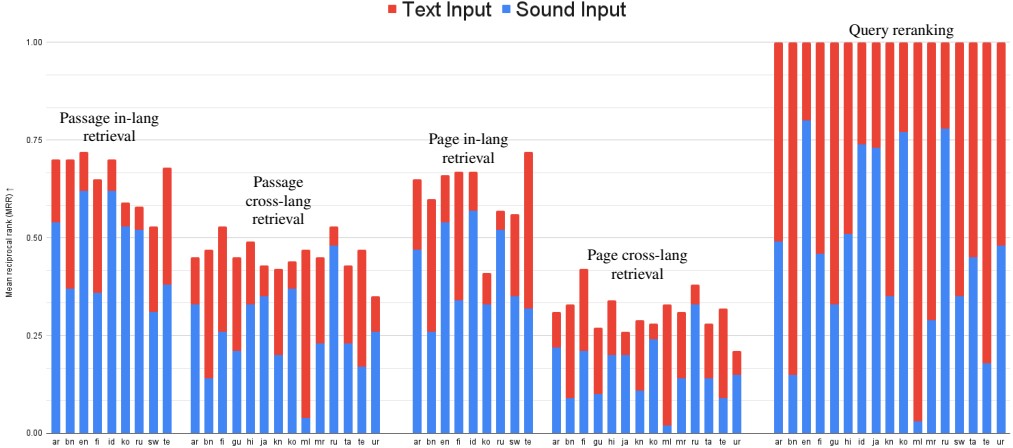

Figure 6: MSEB retrieval and reranking tasks: head room analysis for Gecko.

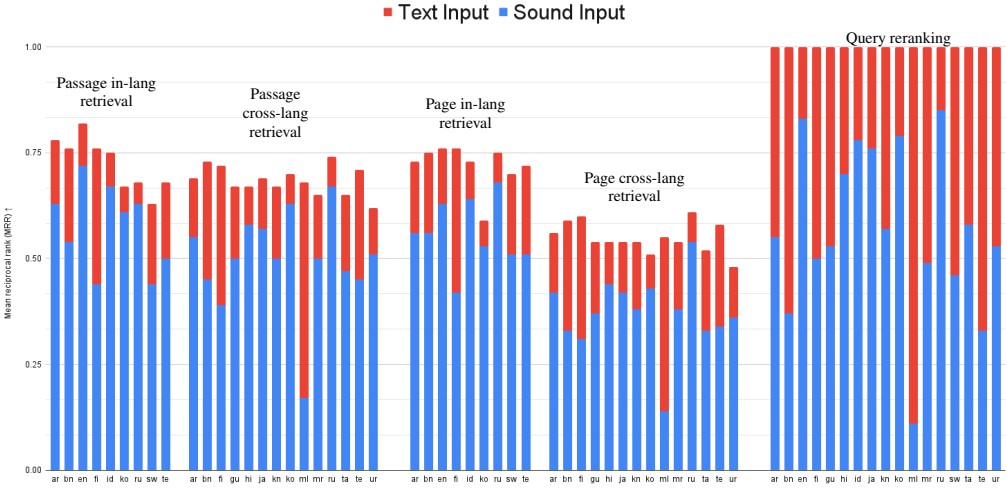

Figure 7: MSEB retrieval and reranking tasks: head room analysis for Gemini Embedding.

## C Reranking

For the query reranking task, our objective is to generate a challenging set of candidate queries. This set is designed to include queries phonetically similar to the ground-truth query, thereby testing acoustic discrimination, and is augmented with semantically similar candidates to serve as plausible distractors to make guessing the answer more difficult. These candidates are initially generated using a large language model (LLM) guided by a prompt that incorporates these phonetic and semantic constraints, see Table 5.

Subsequently, the generated list undergoes a few post-processing steps: addition of the ground-truth query, duplicate removal, sorting by increasing Word Error Rate (WER) relative to the ground-truth query, and truncation to retain up to the top 250 candidates. Given that a single ground-truth query may correspond to multiple recorded utterances (differing in noise conditions or locales), all such utterances are assigned this consistent set of candidate queries. An illustrative example of these generated candidates is presented in Table 6.

Table 5: System instruction templates for generation of phonetically and semantically similar candidates in query reranking task.

Given the sentence '{text}', generate {num_candidates} sentences in {language} that sound phonetically similar to it.
Focus on similar sounding words and rhythm. Do not consider the semantic meaning of the sentences, only the sound.
Output the result as a list, with one sentence per line, without reasoning and comments.

Given the sentence '{text}', generate {num_candidates} sentences in {language} that are semantically similar to it.
Focus on synonyms and similar semantic meaning. Do not consider the sound of the sentences, only the semantic meaning.
Output the result as a list, with one sentence per line, without reasoning and comments.

Table 6: MSEB query reranking candidates for the query "What artists are part of Clamp?"

| | |
|---|---|
| 1. "What artists are part of clamb?" | 9. "What artists are affiliated with Clamp?" |
| 2. "What artists are members of Clamp?" | 10. "What arses are tart of slump?" |
| 3. "What start are part of arm?" | 11. "What actors are start of trump?" |
| 4. "What artists are cut of plane?" | 12. "What arced ships part of sum?" |
| 5. "Which designers are part of Clamp?" | 13. "What artists impart to clamp?" |
| 6. "What designers are a part of Clamp?" | 14. "What farts are part, plum?" |
| 7. "Which animators are part of Clamp?" | 15. "What artists part card swarm?" |
| 8. "What individuals are members of Clamp?" | . . . |

Per-language results for our cascade model using Gemini Embedding are presented in Figure 8. The text-based baseline inherently achieves optimal performance (e.g., MRR=1) in this evaluation because the ground-truth query will always yield the highest similarity score when compared against itself.

We observe that headroom varies across languages. Key observations include:

- Whisper demonstrates significant performance differences across languages (see Fig. 5), with word error rates (WERs) ranging from $10\%$ (en) to $100\%$ (ml).
- Gemini Embedding consistently outperforms Gecko.

Overall, a substantial headroom exists between ASR-based models and ground truth, indicating considerable potential for improved modeling.

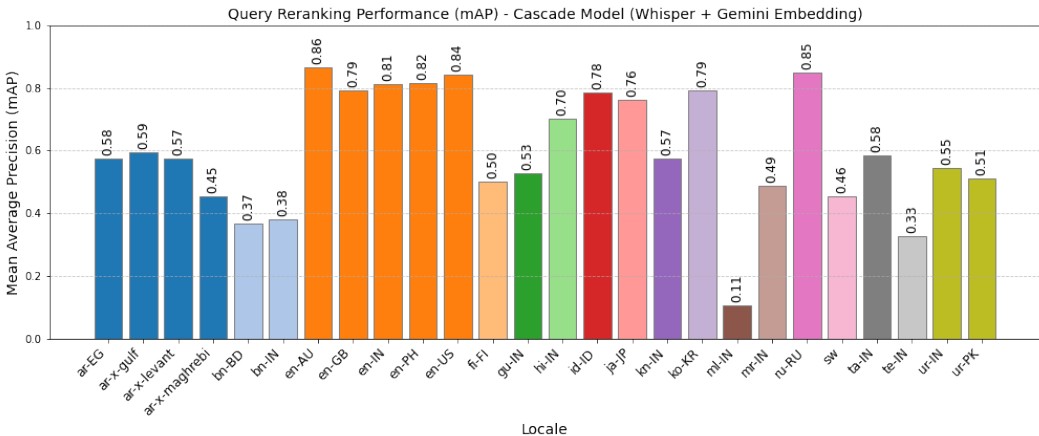

Figure 8: Query Reranking performance (mAP) across SVQ locales using the cascade baseline (Whisper ASR + Gemini Embedding).

# D Reasoning

This paper frames reasoning tasks as follows: given a query and a context, the goal is to predict the corresponding answer span from that context or, if no answer is found, to indicate this. We further distinguish between two types of reasoning tasks based on the context provided: passage-level and page-level. In passage-level reasoning, the context consists of a title and a specific Wikipedia passage, for which we reuse the passages from the original XTREME-UP dataset. For page-level reasoning, the context includes a title and the full Wikipedia page referenced by that title. This wiki-page information is sourced from the wikipedia/20190301 collection.

We provide baseline results for a generative (Gemma 3) and a retrieval (Gemini embedding) approach. In the retrieval approach we use a threshold of 0.8 to discriminate between a real answer and 'No Answer'. The per-language results for these two reasoning methods are presented in Fig. 9. The retrieval approach does not work well, supposedly because both Gemini embedding and Gecko have not been fine-tuned for this type of task.

Table 7: System instruction for reasoning tasks.

---

**Task: Answerability Determination and Exact Answer Extraction**

**Goal:** Determine if a question can be answered using the provided context and title, and if so, extract the **exact** answer verbatim from the text.

**Input:** You will receive a question, title and context each in a new line.

* "question": The question being asked (string).
* "title": The title of the document the context is from (string).
* "context": A text passage which can either be a wiki page or a wiki paragraph that may or may not contain the answer.

**Output:** You will produce a single JSON object as a plain text string (no markup). The structure depends on answerability:

If the question IS answerable:

* "rationale": (string) A concise explanation of why the provided answer is correct. Be specific, referencing sentences or phrases.
* "answer": (string) The answer to the question, copied **exactly** from the title or context. Do not paraphrase or summarize. Prefer concise answers (shortest possible while complete).

If the question IS NOT answerable:

* "No Answer": (string) A clear and concise explanation of why the question cannot be answered. Specify what information is missing.

**Important Considerations:**

* **Code change**: The question may be in a language differ from context and title. In that case, answer the question with the same language as context and title.

* **Exact Matches**: Prioritize using the exact words within the provided text. Do not rephrase or summarize. Do not translate to English.

* **Specificity**: Be as specific as possible in your rationale and no_answer explanations.

* **Title and Context**: Consider both.

* **Direct Answers**: Only use the text from title or context. Do not infer or conclude.

* **Ambiguity**: If a question could have multiple different answers based on the context, the answer should be considered "No Answer" as the context is not specific enough. If multiple answers are equally supported and equally correct, select 'No Answer'.

* **Plain Text JSON Output**: The output must be a valid JSON string, but it must be a plain text string – no markup of any kind.

---

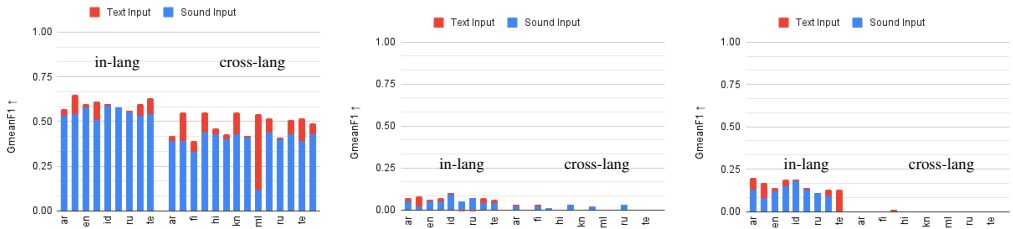

Figure 9: MSEB span reasoning tasks: Generative (Gemma) vs retrieval (Gemini embedding and Gecko) approach.

# E    Classification

We evaluated classification performance across diverse domains, from human speech intent to environmental and bioacoustic sounds.

Table 8 presents reference state-of-the-art benchmarks for intent classification on Speech-MASSIVE and bird-song classification on BirdSet.

For our experiments, we expanded beyond these baselines to include zero-shot and specialized encoder evaluations:

- Intent Classification (Speech-MASSIVE): We evaluated a zero-shot approach using Gemini embedding. As shown in Figure 10, this method achieves strong performance across many languages without task-specific training.

- Bioacoustics (BirdSet): We evaluated Perch, achieving 0.66 mAP on the NBP test set, 0.54 mAP on HSN, and 0.53 mAP on POW.

- Environmental Sounds (FSD50K): We evaluated the CLAP encoder in a zero-shot setting, achieving 0.43 mAP.

Table 8: Reference state-of-the-art classification benchmarks.

| name | | | embedding | | metric | | |
|---|---|---|---|---|---|---|---|
| type | dataset | split | model | type | type | value | reference |
| intent | Speech-MASSIVE | ar
de
es
fr
hu
ko
nl
pl
pt
ru
tr
vi | Whisper | text | acc [%] | 61.22
78.64
80.59
85.93
63.93
74.09
77.49
76.88
80.02
79.51
71.14
68.71 | [14] |
| bird-song | BirdSet | PER
NES
UHH
HSN
NBP
SSW
SNE | wav2vec | vector | cmAP | 0.18
0.39
0.27
0.45
0.63
0.28
0.29 | [16] |

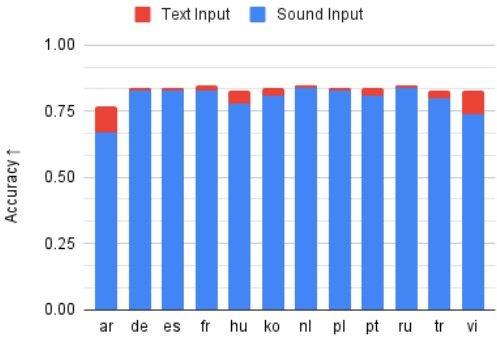

Figure 10: Speech-MASSIVE: Zero-shot intent classification accuracy across languages using Gemini embedding.

# F    Transcription

For the transcription task, we evaluated the Whisper Large v3 model [22] across all 26 language locales available in the SVQ dataset. The evaluation encompassed all four acoustic environments—clean, background speech, traffic noise, and media noise—to assess the model's robustness. Performance was quantified using two standard metrics: Word Error Rate (WER) and Sentence Error Rate (SER).

Figure 11 illustrates the overall WER and SER for each language, averaged across all environments. The results highlight severe performance disparities between languages. While high-resource languages like English (`en`) and Russian (`ru`) achieve excellent performance with low WERs, other languages such as Malayalam (`ml`) and Bengali (`bn`) exhibit extremely high error rates. This underscores the significant headroom remaining for achieving truly universal ASR capabilities.

To further analyze the impact of acoustic conditions, Figures 12 and 13 present the cumulative error rates broken down by environment. As expected, the `clean` condition consistently yields the lowest errors. However, susceptibility to specific noise types varies; for example, `traffic_noise` causes notable degradation in many locales, while others are more heavily impacted by `background_speech`.

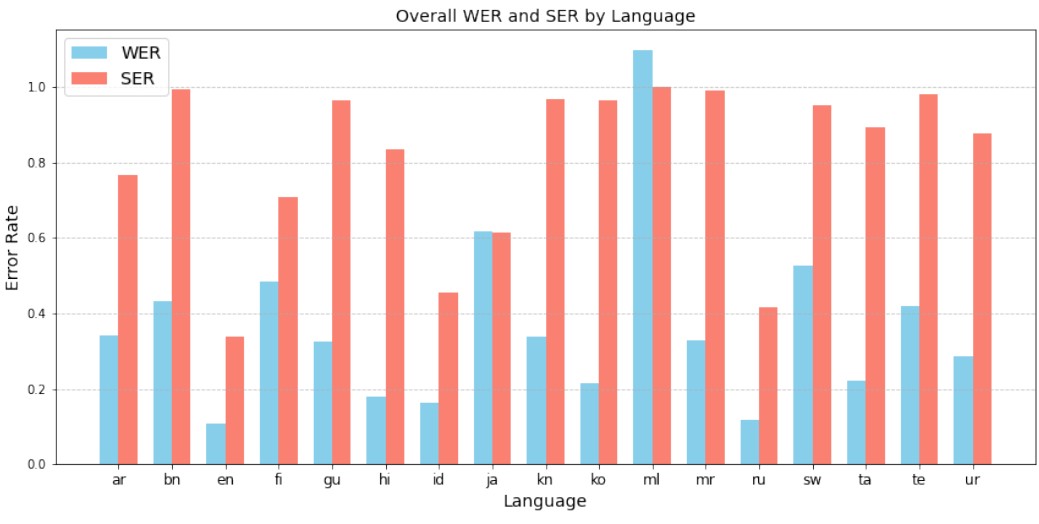

Figure 11: Overall Word Error Rate (WER) and Sentence Error Rate (SER) across all 26 SVQ language locales, averaged over all acoustic environments.

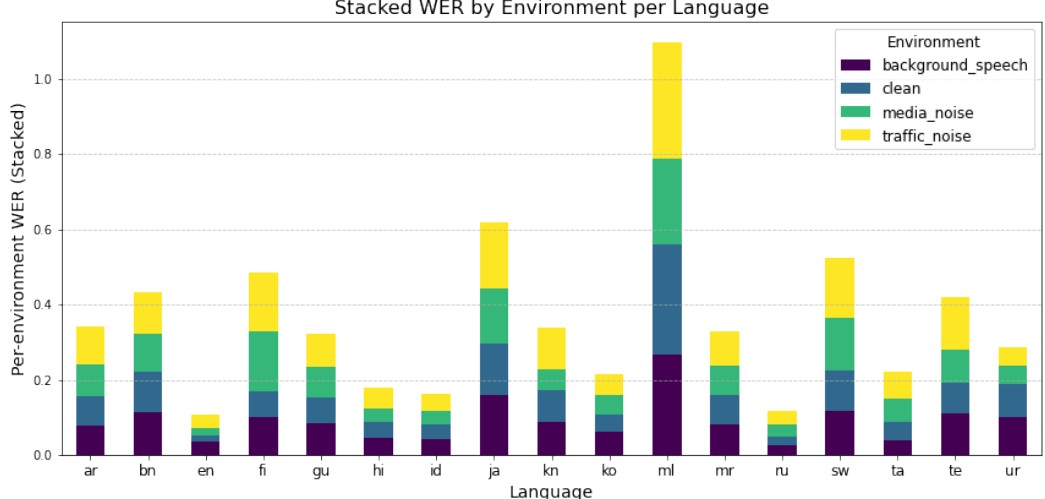

Figure 12: Cumulative Word Error Rate (WER) by environmental condition for each language.

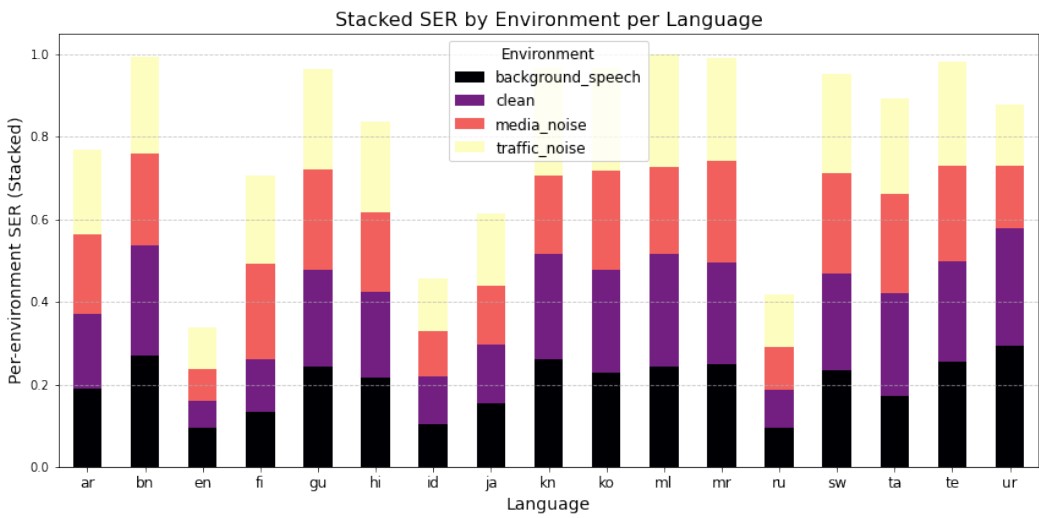

Figure 13: Cumulative Sentence Error Rate (SER) by environmental condition for each language.

# G   Segmentation

For the segmentation task, we employed a cascade baseline designed to locate key information in time. First, the audio query was transcribed using **Whisper Large v3**. Next, we identified the top-three most salient terms in the resulting transcript using a predefined Inverse Document Frequency (IDF) table computed from Wikipedia. Finally, we utilized the word-level timestamps provided by the Whisper model to determine the start and end times for these salient terms.

The primary metric for this task is **Normalized Discounted Cumulative Gain (NDCG)**, which evaluates the quality of the predicted sequence of salient terms and their temporal order. As shown in Figure 14, performance is highly variable across locales. High-resource languages with strong ASR models, such as English (`en-AU`, `en-US`), achieve high NDCG scores (∼0.80). Conversely, languages with poor ASR performance, such as Malayalam (`ml-IN`) and Bengali (`bn-BD`), show near-zero scores.

To understand the sources of error, we broke down performance into three accuracy metrics (Figure 15):

- **Content Accuracy**: Identifying the correct salient term, regardless of timing.
- **Temporal Accuracy**: Identifying the correct time interval, regardless of the term's content.
- **Overall Accuracy (Strict)**: Correctly identifying BOTH the term and its exact time interval.

Figure 15 reveals that the primary bottleneck is often **Content Accuracy**. If the ASR fails to transcribe the salient term correctly (common in difficult languages), strict accuracy naturally falls to zero. Interestingly, for some languages like Korean (`ko-KR`), even when content is identified (high blue bar), temporal alignment remains challenging, leading to a notable drop in strict overall accuracy (green bar).

This strong dependency on ASR quality is confirmed in Figure 16, which shows a clear negative correlation between Word Error Rate (WER) and NDCG.

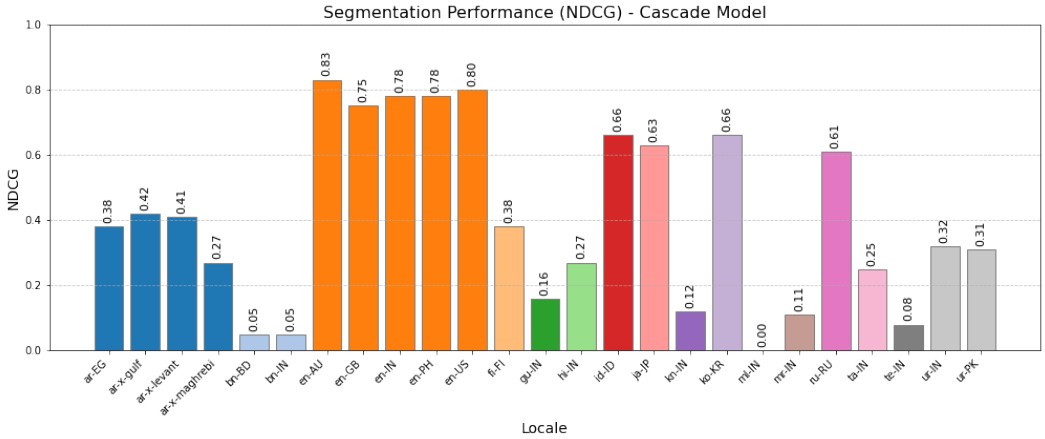

Figure 14: Segmentation performance (NDCG) across all SVQ locales using the cascade baseline.

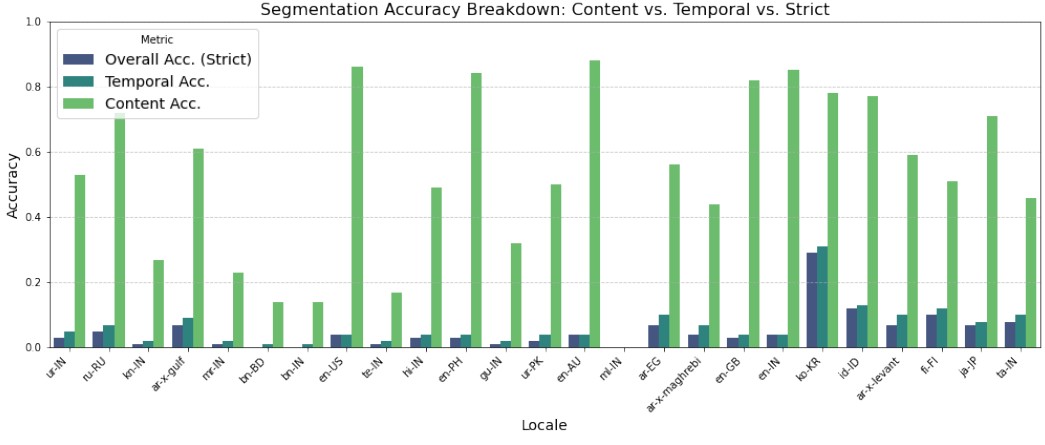

Figure 15: Breakdown of segmentation accuracy. 'Overall Acc. (Strict)' requires both correct content identification and precise temporal localization (within 0.1s tolerance).

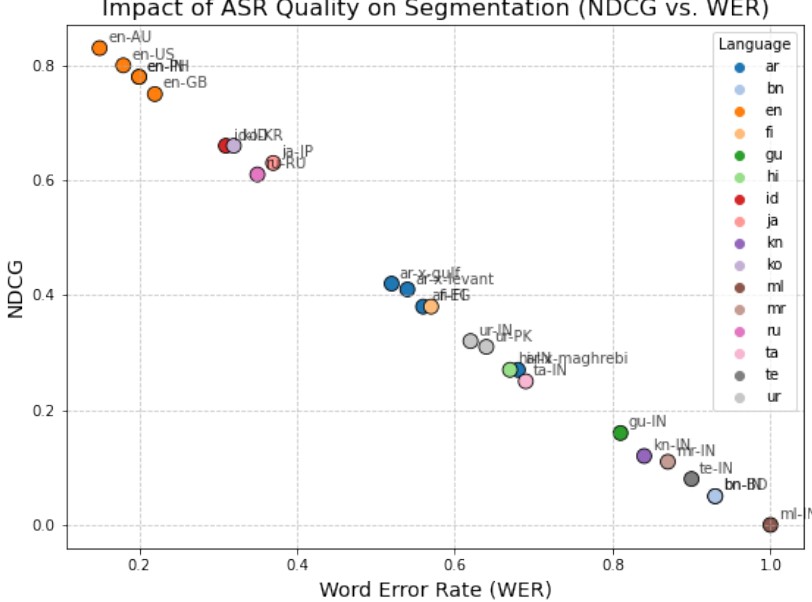

Figure 16: Correlation between ASR quality (WER) and downstream segmentation performance (NDCG), highlighting the cascade model's bottleneck.

## H Clustering

For the clustering super task, we evaluated the capacity of various embeddings to discover latent acoustic structure without supervision. Performance was measured using V-Measure, a standard metric that balances homogeneity and completeness given the ground truth number of clusters.

We conducted experiments across three distinct domains, employing both general-purpose and domain-appropriate encoders:

- **Bioacoustics**: Evaluated on three BirdSet test splits (HSN, NBP, POW) using Perch (specialized for bioacoustics), CLAP (general audio), and a raw Spectrogram baseline.
- **Environmental Sounds**: Evaluated on FSD50K using CLAP.
- **Human Speech**: Evaluated on SVQ across 26 locales for three sub-tasks (Speaker ID, Gender, Age) using self-supervised speech models (HuBERT Large, Wav2Vec2 Large) and a raw Spectrogram baseline.

Figure 17 illustrates performance in non-speech domains. As expected, the specialized Perch model dominates in bioacoustics, achieving V-measures above 0.5 on complex soundscapes (NBP). However, CLAP demonstrates strong value as a generalist encoder, performing respectably on BirdSet while achieving a high V-measure (0.68) on FSD50K environmental sounds.

Figure 18 presents a notable finding in the speech domain. While large pre-trained SSL models (HuBERT, Wav2Vec2) generally perform well, the simple Spectrogram baseline proves highly competitive, particularly for Speaker ID. In many locales, it matches or even exceeds the performance of complex models. This suggests that for the clean, close-talking queries typical of Voice Search (as represented in SVQ), fundamental acoustic features are often sufficient for robust unsupervised speaker discrimination.

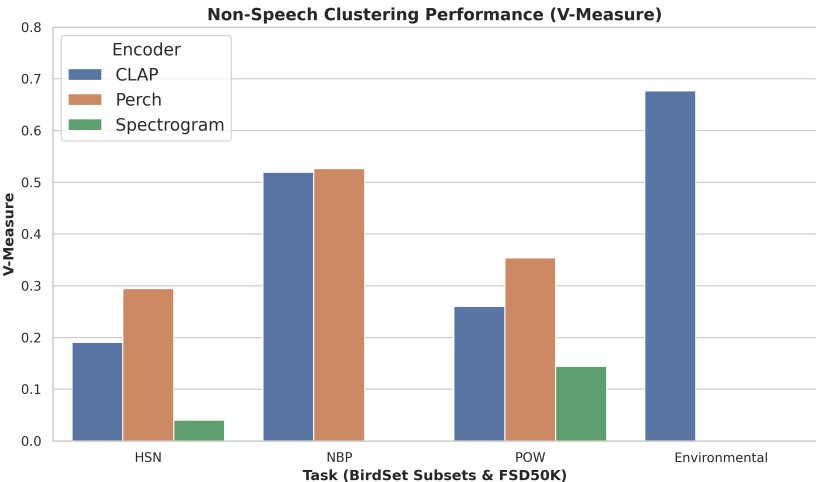

Figure 17: Clustering performance on non-speech domains. Perch excels in its specialized domain of bioacoustics (BirdSet), while CLAP demonstrates strong generalist capabilities across both environmental and bioacoustic tasks.

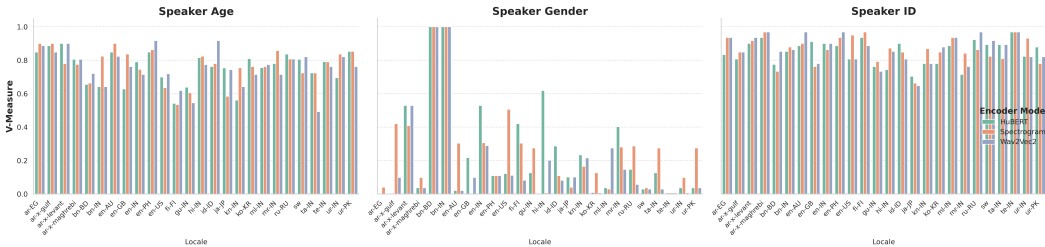

Figure 18: Speaker clustering performance across SVQ locales for Speaker Age, Gender, and ID. Surprisingly, the raw Spectrogram baseline (orange) remains highly competitive with large pre-trained models (HuBERT, Wav2Vec2), especially for Speaker ID.

# I  Reconstruction

For the reconstruction task, we established a baseline using the EnCodec model [30], a widely adopted neural audio codec. The goal was to assess the model's ability to faithfully reconstruct the input audio signal from its compressed representation. We evaluated reconstruction quality using the Fréchet Audio Distance (FAD) [19] metric, calculated by comparing the spectrograms of the original and reconstructed audio. Spectrograms were computed using a frame length of 1024 samples and a frame step of 320 samples.

Experiments were conducted on two distinct domains: speech under various conditions using the SVQ dataset, and general environmental sounds using the FSD50K dataset. The results, summarized in Table 9, reveal significant challenges for current generative audio models.

As shown in Table 9, EnCodec performs best on clean speech, although even here, substantial variation exists across languages, with Hindi (`hi`) showing significantly higher reconstruction error than English (`en`). Performance degrades considerably in the presence of background noise, with traffic noise proving the most detrimental condition. The reconstruction of general environmental sounds from FSD50K yielded the highest FAD score by a large margin, indicating that generating complex, non-speech acoustic scenes remains a particularly difficult task for current models. These results establish a clear baseline and highlight the need for generative models with improved robustness to noise and greater universality across diverse sound types.

Table 9: Reconstruction performance (FAD, lower is better) using EnCodec.

| Dataset | Condition | Mean FAD | Min FAD (Example) | Max FAD (Example) |
|---------|-----------|----------|-------------------|-------------------|
| SVQ | Clean Speech | 103,662 | 15,532 (en) | 193,799 (hi) |
| SVQ | Background Speech | 163,984 | 20,094 (en) | 310,327 (hi) |
| SVQ | Media Noise | 211,774 | 125,050 (ko) | 346,577 (hi) |
| SVQ | Traffic Noise | 348,885 | 133,411 (ko) | 596,491 (hi) |
| FSD50K | Environmental Sound | 778,232 | N/A | |

