# OpenReview forum: "Massive Sound Embedding Benchmark (MSEB)"
_NeurIPS.cc/2025/Datasets_and_Benchmarks_Track — NeurIPS 2025 Datasets and Benchmarks Track poster_

### Official Review · Reviewer_FZ35 · 2025-06-10

**Rating:** 5
**Confidence:** 3

**Summary:**

The Massive Sound Embedding Benchmark (MSEB) is introduced as a new, standardized tool for evaluating how well AI models can extract information from various types of audio. It includes tasks and datasets covering diverse applications like voice search, assistants, monitoring, and bioacoustics, addressing a gap in existing sound benchmarks by focusing on real-world technological problems. The benchmark categorizes sound information extraction into discrete or continuous embeddings, analyzing their compression rates and computational complexities. It includes tasks like retrieval, reasoning, classification, segmentation, clustering, and reranking, with initial results showing significant room for improvement in current methods.

**Dataset Code Accessibility:**

Yes

**Ethical Considerations:**

No, there are no or only very minor ethics concerns

**Limitations Weaknesses:**

The experiments showcase the benchmark's utility by evaluating existing models for specific tasks (e.g., Whisper for ASR, Wav2vec for bird-song). However, the initial results primarily highlight "headroom" for improvement in existing methods  rather than thoroughly demonstrating or analyzing the performance of a single, truly universal sound embedding across all diverse tasks within the benchmark's framework.

**Strengths Contributions:**

The Massive Sound Embedding Benchmark (MSEB) is a new, standardized benchmark for evaluating sound information extraction. It provides a unified framework to assess the optimality of sound representations across diverse applications and sound types.

This benchmark encompasses realistic tasks and datasets reflecting practical applications in areas like voice search, assistants, monitoring, and bioacoustics, and is publicly available to encourage community contributions.

The benchmark introduces novel evaluation criteria, including sound embedding compression ratio (SECR) and computational complexity (FLOPS), and avoids fine-tuning during evaluation, differentiating it from prior benchmarks.

---

> ### Author Rebuttal · Authors · 2025-07-31
>
> We are extremely grateful for your positive evaluation and "Accept" recommendation for our paper. We especially appreciate your clear and insightful summary of our contributions, such as recognizing that MSEB *"provides a unified framework to assess the optimality of sound representations across diverse applications and sound types."* Thank you for your strong support.
>
> Before addressing your specific point, we find it helpful to briefly reiterate the three primary goals of our work, which we will emphasize more clearly in the revised manuscript's introduction. Our motivation for creating MSEB was to:
>
> 1. Introduce and standardize evaluation for a diverse set of crucial, **real-world sound-centric tasks**: retrieval, reasoning, classification, segmentation, clustering, and reranking. Among these, tasks like **retrieval, reasoning, and reranking** have been less visited in the open-source benchmark space.
> 2. Provide the research community with a benchmark that is **maximally easy to use and extend**, allowing any researcher to seamlessly integrate their own embedding algorithms and evaluate them across a wide array of tasks.
> 3. **Establish clear performance headrooms** for these tasks, demonstrating the significant research opportunities that exist beyond conventional cascade-based approaches.
>
> We believe that a paper based on these desiderata aligns well with the scope of the NeurIPS Datasets & Benchmarks Track. To clarify the scope of this work, our focus was on building the benchmark framework itself rather than publishing an exhaustive evaluation of all possible encoders within this single paper. The goal is to provide the community with a robust and easy-to-use codebase to evaluate any encoder. Similarly, while the long-term vision is to enable the development of universal sound embeddings, the scope of this paper is to provide the foundational first step—a standardized and comprehensive benchmark—to help the community move toward that goal.
>
> We will now address the point you raised in your review.
>
> > "The experiments showcase the benchmark's utility by evaluating existing models for specific tasks... However, the initial results primarily highlight "headroom" for improvement in existing methods rather than thoroughly demonstrating or analyzing the performance of a single, truly universal sound embedding across all diverse tasks within the benchmark's framework."
>
> This is an excellent point, as it perfectly separates the primary goals of this paper from the ultimate goal of our research field. As you noted, our work focused on establishing performance headrooms. This aligns with our **primary goal**: to provide the community with a standardized set of tasks, a robust evaluation framework, and strong baselines. The creation of a single universal model, on the other hand, represents the **ultimate goal**—one we hope to achieve in collaboration with the research community—which is to foster the development of a sound representation that excels across these diverse tasks and closes the performance gaps we have highlighted.
>
> To the best of our knowledge, such a universal model does not exist today. Our work, therefore, aims to provide the means for the community to build one. We believe this requires three foundational components, all of which are contributions of this paper. First, it is necessary to **establish a diverse set of important sound-centric tasks**, which we believe MSEB accomplishes with its super tasks. Second, a **robust evaluation framework** is needed, which we provide with the open-source MSEB library. Finally, **strong headroom baselines** must be set to reveal the rich research opportunities available, a goal our evaluation achieves by using state-of-the-art models for ASR (Whisper v3) and text embedding (Gemini Embedding).
>
> By providing these pillars, MSEB is designed to empower the community to take on the challenge of creating a truly universal model. We will be sure to frame this more explicitly in our conclusion.
>
> Thank you once again for your strong support and for your thoughtful assessment of our work.

---

> > ### Comment · Reviewer_FZ35 · 2025-08-05
> >
> > I would like to thank authors for their thoughtful rebuttal. Most of my concerns have been addressed, and I would like to keep my original rating as Accept.

---

### Official Review · Reviewer_zujg · 2025-07-02

**Rating:** 4
**Confidence:** 2

**Summary:**

The paper presents the Massive Sound Embedding Benchmark (MSEB), whose evaluation strategy does not involve any fine-tuning step.
It also introduces the Simple Voice Questions (SVQ) dataset for voice search and reasoning tasks.
However, the benchmark design lacks a broader selection from various datasets and comparison methods.
The presentation is not sophisticated.

(Note: after the rebuttal, the concerns of benchmark design, experimental evaluation, and presentation are solved.)

**Dataset Code Accessibility:**

Yes

**Ethical Considerations:**

No, there are no or only very minor ethics concerns

**Final Justification:**

The rebuttal solves the concerns of benchmark design, experimental evaluation, and presentation.
I raise the score.

**Limitations Weaknesses:**

* The benchmark design lacks a broader selection: In Section 2.2, most of the tasks are based on the SVQ dataset. Other datasets are only used for classification tasks.
* The paper lacks comparison methods and experimental results: In Figure 3 and Table 3, the paper only tested Gecko and Gemini embedding. Table 2 only displays the reference results from the cited sources.
* The presentation is not sophisticated, as shown below.
  - BirdSet has no reference in the paper: "BirdSet: A Large-Scale Dataset for Audio Classification in Avian Bioacoustics"?
  - Compared to the results section, the dataset description is relatively long. Since most of the datasets are not introduced in this paper, the description can be shorter. Instead, the paper can use a longer results section.
  - FSC22 (on page 6) is not defined.
  - [Figure ???] (on page 8) is not defined.
  - In Table 2, BirdSet cites [31]. But [31] is the FSD50K paper.
  - In table 3, "0.00 in Reference \to Gecko" is redundant.

(Note: after the rebuttal, the concerns of benchmark design, experimental evaluation, and presentation are solved.)

**Strengths Contributions:**

* The paper presents the Massive Sound Embedding Benchmark (MSEB)
  - it selects datasets and tasks that address problems with wide applicability
  - its evaluation strategy does not involve any fine-tuning step
* The paper introduces the Simple Voice Questions (SVQ) dataset for voice search and reasoning tasks
  - it comprises 177,352 audio queries recorded in 26 locales across 17 languages

---

> ### Author Rebuttal · Authors · 2025-07-31
>
> We would like to thank you for your time and for providing a detailed review of our work. We appreciate that you recognized the key strengths of our contribution, such as selecting datasets and tasks with wide applicability and employing an evaluation strategy that does not require fine-tuning. Your critical feedback is invaluable, and we have a clear plan to address the weaknesses you identified to improve the paper.
>
> Before addressing your specific points, we find it helpful to briefly reiterate the three primary goals of our work, which we will emphasize more clearly in the revised manuscript's introduction. Our motivation for creating MSEB was to:
>
> - Introduce and standardize evaluation for a diverse set of crucial, **real-world sound-centric tasks**: retrieval, reasoning, classification, segmentation, clustering, and reranking. Among these, tasks like **retrieval, reasoning, and reranking** have been less visited in the open-source benchmark space.
> - Provide the research community with a benchmark that is **maximally easy to use and extend**, allowing any researcher to seamlessly integrate their own embedding algorithms and evaluate them across a wide array of tasks.
> - **Establish clear performance headrooms** for these tasks, demonstrating the significant research opportunities that exist beyond conventional cascade-based approaches.
>
> We believe that a paper based on these desiderata aligns well with the scope of the NeurIPS Datasets & Benchmarks Track. To clarify the scope of this work, our focus was on building the benchmark framework itself rather than publishing an exhaustive evaluation of all possible encoders within this single paper. The goal is to provide the community with a robust and easy-to-use codebase to evaluate any encoder. Similarly, while the long-term vision is to enable the development of universal sound embeddings, the scope of this paper is to provide the foundational first step—a standardized and comprehensive benchmark—to help the community move toward that goal.
>
> We will now address your specific concerns.
>
> > "The benchmark design lacks a broader selection: In Section 2.2, most of the tasks are based on the SVQ dataset. Other datasets are only used for classification tasks."
>
> Your point gives us an opportunity to explain our two-pronged design philosophy, which we believe is a key strength of our benchmark.
>
> First, to the best of our knowledge, no other single public dataset exists that can support the full range of important, real-world tasks we introduce, especially complex ones like retrieval, reasoning, and reranking. To address this critical gap, we created the SVQ dataset. With its comprehensive coverage—spanning 17 languages, 27 locales, 4 environments, and rich metadata including speaker ID, gender, age, and ground-truth page, passage, and span answers—it provides the first unified foundation for these tasks.
>
> Second, while SVQ provides breadth, we deliberately included well-established, single-task datasets for depth and comparability. Classification, for example, is a mature field with excellent datasets. A truly universal embedding, which MSEB aims to foster, should not only perform well on broad tasks but must also prove its capabilities on these specialized, well-understood domains. Including them is essential for robustly evaluating any claims of universality.
>
> Therefore, the inclusion of both the purpose-built SVQ and established classification datasets is a core feature of MSEB's design, enabling a more holistic and rigorous evaluation than either type of dataset could provide alone. We will ensure this rationale is more prominent in our revision.
>
> > "The paper lacks comparison methods and experimental results: In Figure 3 and Table 3, the paper only tested Gecko and Gemini embedding. Table 2 only displays the reference results from the cited sources."
>
> Our primary goal for the initial experiments was to establish a meaningful performance headroom for the widely-used cascade modeling approach. To do this, we selected state-of-the-art components for each stage: we used Whisper Large v3, as it represents the state-of-the-art in ASR across many languages, and Gemini Embedding, which is a top-performing text embedding model on benchmarks like MTEB. This allowed us to create a highly competitive baseline to measure against. In the revised version of the paper, we will expand our evaluation to include other important self-supervised models, such as HuBERT and WavLM, to provide a broader set of comparison points.
>
> Regarding the cited results in Table 2, our initial approach was to cite the original published numbers for fairness. However, we agree with your sentiment that a self-contained benchmark is stronger. In the final version of the paper, we will include results from our own reproduction experiments for these models to provide a direct and consistent comparison point for the community.
>
> > "Compared to the results section, the dataset description is relatively long... the description can be shorter. Instead, the paper can use a longer results section."
>
> This is an excellent suggestion for improving the paper's structure and impact. We agree completely. We would like to clarify that our initial submission already contains a substantial number of results and comparisons, which are detailed across **eight appendices**, with each super-task (retrieval, reasoning, classification, segmentation, clustering, and reranking) having its own dedicated appendix. We recognize that these extensive results may not be immediately visible in their current location. Therefore, in our revision, we will adopt your suggestion to condense the dataset descriptions in the main body. We will then use the newly available space to provide a comprehensive summary of the key findings from our appendices, better highlighting the breadth and depth of our evaluation.
>
> We thank you for your careful attention to detail and will correct all the presentation issues you have identified. We are confident that by addressing these points, we can significantly improve the paper and better convey the value and utility of the MSEB framework.

---

> > ### Comment · Reviewer_zujg · 2025-08-03
> > **Response to Rebuttal**
> >
> > I thank the authors for the careful rebuttal.
> >
> > I understand that the core design of MSEB is based on the use of both the purpose-built SVQ and established classification datasets.
> > It is appreciated to emphasize this point in the revision.
> > The SVQ dataset is a considerable contribution of the paper.
> >
> > It is also appreciated to include other self-supervised models in the SVQ experiments and include the reproduction results in Table 2.
> >
> > The core parts of the SVQ dataset should be explained in the main body since it is introduced in this paper.
> > On the other hand, the explanation of the other datasets can be summarized in the main body.
> > I thank the authors for improving the main body with the key results from the eight appendices.
> > That would help future readers to grasp the overview using the main body.

---

> > > ### Author Response · Authors · 2025-08-04
> > >
> > > Thank you for your constructive review. We're pleased our response addressed your concerns and will incorporate all discussed revisions into the final manuscript.

---

### Official Review · Reviewer_3VYV · 2025-07-03

**Rating:** 5
**Confidence:** 3

**Summary:**

The authors propose the Massive Sound Embedding Benchmark (MSEB), a collection of datasets (SVQ, Speech-MASSIVE, TUT18, FSD50K, BirdSet) and evaluation tasks (retrieval, reasoning, classification, segmentation, clustering, and reranking) to evaluate sound embeddings in realistic and practical. Moreover, the authors benchmarked SoTA models (Whisper Large v3 + Gemini/Gecko Embedding, etc.) on their dataset, revealing significant headroom for improvement in current methodologies.

**Additional Feedback:**

1. There is a missing **and** in "There are also speech-specific benchmarks like the non-semantic speech benchmark (NOSS) [20] **and** speech processing universal performance benchmark (SUPERB) [21]".
1. "The query’s text comes from validation and test sets of the XTREME-UP’s retrieval and question answering **benchamark** datasets [23]" should be **benchmark**.
1. Missing reference in "The retrieval quality, measured by Mean Reciprocal Rank (MRR), is presented in Figure **[Figure ???]**."
1. The figures are low-resolution bitmaps, but they could preferably be more crisp vector graphics.

**Dataset Code Accessibility:**

Partly

**Dataset Code Comments:**

The SVQ dataset is available at https://huggingface.co/datasets/google/svq, and the source code is available at https://github.com/google-research/mseb. However, little instruction/documentation is given on how to replicate the experiments.

**Ethical Considerations:**

No, there are no or only very minor ethics concerns

**Final Justification:**

I'm increasing the score because the authors are reproducing experiments from other papers. I really appreciate that because reproducibility is a big deal, specially for a paper in the Datasets and Benchmarks Track. I'm also glad to hear that the authors are including HuBERT and WavLM in their benchmark. It's not trivial for a framework to be widely adopted by the relevant community, so I hope the authors put continuous efforts into preaching about the MSEB framework and make it more extensible/adaptable.

**Limitations Weaknesses:**

1. The only "novel" dataset appears to be Simple Voice Questions (SVQ), whereas Speech-MASSIVE, TUT18, and FSD50K all come from prior literature.
1. The benchmark only evaluates a small number of sound embedding, i.e. wav2vec and cascade modeling with Whisper Large v3 + Gemini/Gecko Embedding. The authors should include more recent developments like [HuBERT](https://arxiv.org/abs/2106.07447) and [WavLM](https://arxiv.org/abs/2110.13900).
1. For some experiments, the authors are citing some numbers from other papers instead of running the experiments on their own, which causes the experiment results to be quite minimal. For example, Table 2 does not tell us the performance of wav2vec on intent classification on Speech-MASSIVE.

**Strengths Contributions:**

The authors outlines the contribution of their work as follows, which I mostly agree.
> MSEB differentiates itself from these prior benchmarks in four key ways: 1) its core design principle prioritizes selecting datasets and tasks that address problems with wide applicability in technology or the scientific community. 2) it emphasizes measuring the compression ratio and computational complexity of the embeddings. 3) it includes datasets and tasks, such as retrieval and reasoning, which represent important speech use cases not covered in previous benchmarks. 4) unlike benchmarks such as NOSS, SUPERB, or HARES, which require fine-tuning a shallow neural network for each task on top of frozen embeddings during evaluation, MSEB’s evaluation methodology does not involve any fine-tuning step.

Moreover, all included datasets are open access, ensuring the reliability of longitudinal comparisons.

On the other hand, the evaluation tasks and benchmark reveal substantial headroom for enhancing prevalent information extraction methodologies. Among others, multiple experiments in Section 3 indicate that the conventional cascade modeling approach for voice search still has substantial room for improvement.

---

> ### Author Rebuttal · Authors · 2025-07-31
>
> We wish to extend our sincerest gratitude for your insightful review and constructive feedback. We are very encouraged by your "borderline accept" rating and that you mostly agree with the main contributions of our work, particularly the four key ways MSEB differentiates itself from prior benchmarks. Your feedback has provided a clear roadmap for significantly improving our paper.
>
> Before addressing your specific points, we find it helpful to briefly reiterate the three primary goals of our work, which we will emphasize more clearly in the revised manuscript's introduction. Our motivation for creating MSEB was to:
> - Introduce and standardize evaluation for a diverse set of crucial, **real-world sound-centric tasks**: retrieval, reasoning, classification, segmentation, clustering, and reranking. Among these, tasks like **retrieval, reasoning, and reranking** have been less visited in the open-source benchmark space.
> - Provide the research community with a benchmark that is **maximally easy to use and extend** allowing any researcher to seamlessly integrate their own embedding algorithms and evaluate them across a wide array of tasks.
> - **Establish clear performance headrooms** for these tasks, demonstrating the significant research opportunities that exist beyond conventional cascade-based approaches.
>
> We believe that a paper based on these desiderata aligns well with the scope of the NeurIPS Datasets & Benchmarks Track. To clarify the scope of this work, our focus was on building the benchmark framework itself rather than publishing an exhaustive evaluation of all possible encoders within this single paper. The goal is to provide the community with a robust and easy-to-use codebase to evaluate any encoder. Similarly, while the long-term vision is to enable the development of universal sound embeddings, the scope of this paper is to provide the foundational first step—a standardized and comprehensive benchmark—to help the community move toward that goal.
>
> We will now address the specific concerns and feedback you raised.
>
> > "The only 'novel' dataset appears to be Simple Voice Questions (SVQ), whereas Speech-MASSIVE, TUT18, and FSD50K all come from prior literature."
>
> We'd like to emphasize that SVQ is a comprehensive 'super-dataset'. Beyond its wide coverage of 26 locales, 17 languages, and 4 recording environments, each example is enriched with detailed metadata, including speaker ID, gender, age, and ground-truth answers for pages, passages, spans, and ground-truth transcription. This rich metadata provides the precise evaluation basis for every task in our benchmark. Furthermore, SVQ is a dynamic resource that is actively growing to include more languages and diverse multimodal queries. We will clarify these points in the revised manuscript.
>
> > "The benchmark only evaluates a small number of sound embedding, i.e. wav2vec and cascade modeling with Whisper Large v3 + Gemini/Gecko Embedding. The authors should include more recent developments like HuBERT and WavLM."
>
> Thank you for the excellent suggestion. We will make sure to include both HuBERT and WavLM in the final version of the paper. It's important to note that MSEB is designed as a dynamic and growing benchmark. A core feature of our framework is its easy extensibility, allowing for the seamless integration of any new encoder model. While we will include these models in our revision, we also wish to strongly encourage submissions from the community, as our framework is designed to make evaluating any existing and new model a straightforward process.
>
> > "For some experiments, the authors are citing some numbers from other papers instead of running the experiments on their own, which causes the experiment results to be quite minimal. For example, Table 2 does not tell us the performance of wav2vec on intent classification on Speech-MASSIVE."
>
> This is a very thoughtful point. We have been working hard to reproduce the published results for these open-source models on their respective tasks. In this process, we assumed that the most transparent and fair approach to representing the state-of-the-art performance for those specific model-task pairs was to cite their officially published numbers. However, we agree that providing our own results is valuable for a self-contained benchmark. In the final version of the paper, we can add the results from our own reproduction experiments alongside the cited numbers for full transparency.
>
> > "little instruction/documentation is given on how to replicate the experiments."
>
> Thank you for highlighting this. We agree that full reproducibility is paramount. To address this, we will enhance our GitHub repository with a comprehensive README.md and supplementary documentation. This will include step-by-step instructions, environment details, and the exact command-line examples required to reproduce every table and figure in our paper. Furthermore, we will add a new appendix detailing the submission process for the MSEB benchmark.
>
> Thank you as well for your meticulous review; we will be sure to incorporate all the corrections you suggested in the 'Additional Feedback' section.

---

> ### Comment · Reviewer_3VYV · 2025-08-06
>
> Thanks for the detailed rebuttal. Most of my concerns have been addressed. I did have some misunderstanding regarding the composition of Simple Voice Questions (SVQ), and I'm glad that the authors can clarify that for me.
>
> I'm increasing the score because the authors are reproducing experiments from other papers. I really appreciate that because reproducibility is a big deal, specially for a paper in the Datasets and Benchmarks Track. I'm also glad to hear that the authors are including HuBERT and WavLM in their benchmark. It's not trivial for a framework to be widely adopted by the relevant community, so I hope the authors put continuous efforts into preaching about the MSEB framework and make it more extensible/adaptable.

---

### Note · Authors · 2025-08-13

We wish to sincerely thank all three reviewers and the Area Chair for their valuable time and for a very productive discussion period. We are delighted that our rebuttal has addressed their concerns.
We fully commit to implementing all the promised revisions in the final manuscript, including the addition of new models, the reproduction of cited results, and the restructuring of the paper to improve clarity and impact.
Thank you again for your consideration and guidance.

---

### Decision · Program_Chairs · 2025-09-18

**Decision:**

Accept (poster)

**Comment:**

This paper presents MSEB (short for Massive Sound Embedding Benchmark), which is a unified framework with a collection of datasets and evaluation tasks in both retrieval and reasoning. This preliminary version is focused on sound-based tasks while the authors also mention future plan to incorporate speech-centric tasks, music tasks, and other dataset types. Importantly, MSEB doesn't involve any finetuning for each evaluation tasks.

There are also several initial concerns/weaknesses of this paper raised by the reviewer, primarily centered around benchmark design (SVQ vs. other datasets, more sound embeddings), experimental evaluation (reproducing own results vs. citing numbers in references, documentation for reproducibility), and presentation (broken references, more clarifications on two-pronged design philosophy, existence of a universal model/embedding space), all of them are well addressed during rebuttal. Thus, I am recommending acceptance.

===== FINAL UPDATE FROM DB Track PCs ====

The final decision for this paper has been taken by the program chairs after consultation with the SACs. All Senior Area Chairs have ranked papers according to the feedback from the AC during the review process. We decided to leave the original meta-review to reflect the opinion of the AC in light of the initial discussions with reviewers and SAC.